# Warmer environments harbor greater thermal trait diversity in moth assemblages

Ming Liu[1,2,10], Tzu-Man Hung [1,3,4,10], Shipher Wu [1,5,10], Mark Liu[1], Guan-Shuo Mai [1], Yi-Shin Jang[1,6], Chien-Chen Huang[4,7], Chun-Yung Hsu[1,8,9], Chia-Hsuan Wei [4], Mao-Ning Tuanmu [1], Shih-Fan Chan [1], I-Ching Chen [4] ✉ & Sheng-Feng Shen [1,3,8] ✉

Thermal trait diversity is critical for understanding species' responses to climate change, yet its ecological drivers remain unclear. Using eco-evolutionary simulations and empirical data from 653 moth species across three Asian elevational gradients, we examine how temperature regimes shape thermal strategies in assemblages. Warmer environments support larger hypervolumes of moth assemblages, reflecting a broader array of coexisting thermal strategies. Contrary to the climatic variability hypothesis, which predicts generalized traits under stable climates, we find that warmer sites foster assemblage-level diversity even while individual species retain narrow thermal tolerance ranges. Short-term temperature fluctuations exert minimal influence, while seasonal variability promotes generalists but reduces overall hypervolume. These results demonstrate that mean temperature, not variability, is the dominant force structuring thermal trait diversity. By revealing how thermal strategies assemble under different climates, our study provides a mechanistic basis for predicting biodiversity responses to warming and emphasizes the conservation value of low-elevation ecosystems.

Understanding trait diversity has been central to ecology since Humboldt, Darwin, and Wallace first documented tropical organisms' remarkable variation in form and function[1–4]. This early recognition of trait diversity's importance has evolved into modern approaches examining both species and assemblage levels. While species-level studies provide foundational insights, assemblage-level research has emerged as crucial for understanding physiological trait variation[5]. The strength of assemblage-level analyses is that they capture emergent properties—attributes of the whole assemblage that cannot be inferred by inspecting species in isolation. For example, coral assemblages recover only 60% of their original trait space after extreme heat waves[6], a pattern undetectable through individual species analyses[7–9].

However, conducting large-scale studies of assemblage-level physiological traits presents significant challenges, particularly in data collection across broad environmental gradients. These challenges have historically limited our understanding of how environmental conditions shape thermal adaptations at the assemblage level.

Three alternative hypotheses have been proposed to explain thermal trait patterns across environmental gradients, each offering distinct and testable predictions. These mechanisms—favorability, long-term seasonal variation, and short-term daily fluctuation—are not mutually exclusive and may jointly influence trait evolution depending on the ecological context and temporal scale. The favorability hypothesis, developed from Wallace's 1878 observations, proposes

[1]Biodiversity Research Center, Academia Sinica, Taipei, Taiwan, ROC. [2]Department of Biology, University of Oxford, Oxford, UK. [3]Institute of Ecology and Evolutionary Biology, National Taiwan University, Taipei, Taiwan, ROC. [4]Department of Life Sciences, National Cheng Kung University, Tainan, Taiwan, ROC. [5]National Taiwan Museum, Taipei, Taiwan, ROC. [6]Department of Atmospheric Sciences, National Taiwan University, Taipei, Taiwan, ROC. [7]School of Natural Sciences, Macquarie University, Sydney, NSW, Australia. [8]International Degree Program in Climate Change and Sustainable Development, National Taiwan University, Taipei, Taiwan, ROC. [9]Social Science Research Promotion Project for Taiwan Net-Zero Pathway, National Science and Technology Council, Taipei, Taiwan, ROC. [10]These authors contributed equally: Ming Liu, Tzu-Man Hung, Shipher Wu. ✉e-mail: chenic@nckU.edu.tw; shensf@sinica.edu.tw

that benign tropical environments permit continuous evolution of life forms under reduced selective pressures[10] (Fig. 1a). Fischer extended this concept, predicting higher functional diversity in favorable tropical environments, which aligns with both the "more-individuals hypothesis"[5] and "hotter is better" principle[11–14]. These interconnected theories suggest that favorable conditions promote not only greater abundance and species richness but also broader functional trait diversity through reduced environmental filtering in productive environments. The mechanism underlying this pattern involves complex interactions between environmental productivity and species'

thermal adaptations, allowing the persistence of diverse thermal strategies.

In contrast, Janzen[15] proposed that species in environments with lower seasonal temperature variations evolve narrower thermal tolerances, explaining tropical biodiversity patterns through climatic stability[15–17]. This perspective spawned Rapoport's rule, which suggests that decreasing seasonal variation at lower latitudes leads to reduced geographical ranges and higher species overlap[18] (Fig. 1b). The implications of this climatic variability hypothesis (also referred to as the long-term variability hypothesis) extend beyond species distributions

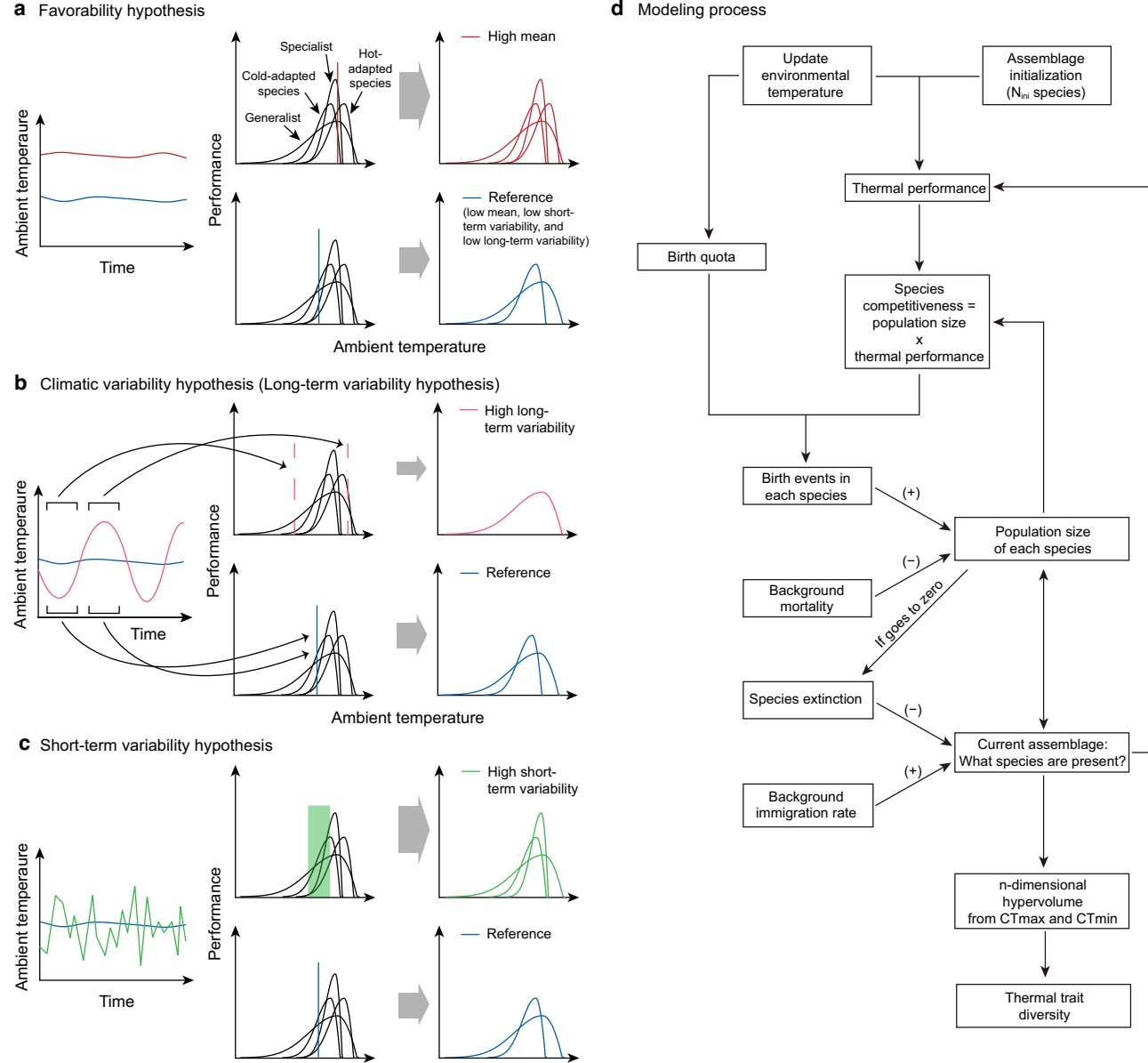

**Fig. 1 | Summary of the favorability, climatic variability hypotheses, and short-term variability and their key predictions regarding thermal trait diversity at the assemblage level.** Each panel shows (1) an environmental time series in the left column, (2) the assemblage before environmental filtering with summarized environmental conditions in the center column, and (3) the surviving species after environmental filtering in the right column. **a** The favorability hypothesis predicts that thermal traits within an assemblage (the set of species in a single place) are more diverse in warmer places, due to greater productivity. **b** The climatic variability hypothesis predicts that thermal trait diversity decreases with climatic variability (usually referred to as seasonality). **c** The short-

term variability hypothesis predicts that thermal trait diversity increases with short-term variability. **d** Schematic overview of the process in the eco-evolutionary model. After initialization, each time step repeats the process from updating environmental temperature to the current assemblage. The arrows from population size and assemblage back to thermal performance and species competitiveness indicate the information from the previous time step is used as input (e.g., whether we need to consider a new thermal performance because a species migrated in the last time step). The calculation of n-dimensional hypervolume is only carried out after the simulation is completed (i.e., final assemblage).

to fundamental questions about the evolution of thermal tolerance and its relationship with environmental variability. Nevertheless, empirical support for this hypothesis has been mixed, particularly regarding its predictions about thermal trait evolution across different taxonomic groups and geographic regions[19–21].

Recent work has introduced an alternative perspective focusing on short-term environmental variability[16], suggesting that daily temperature fluctuations favor thermal specialization by providing frequent favorable conditions[16,22,23] (Fig. 1c). Short-term temperature variability is thought to enhance trait diversity in a manner fundamentally different from long-term seasonal variation. This hypothesis provides a framework for understanding how different temporal scales of environmental variation might select for different thermal strategies. The distinction between short-term and long-term variability effects represents an important advance in our understanding of thermal adaptation mechanisms, because it highlights that environmental fluctuations at different temporal scales can select for fundamentally different thermal strategies. While long-term variability tends to favor generalists with broad tolerance ranges, short-term fluctuations may support specialists adapted to transient favorable conditions[16,22,23]. Recognizing this separation allows for more precise predictions about how organisms respond to climate dynamics, especially in the face of increasing environmental variability under global change.

Although adaptation to the environment at the species level is closely related to the diversity of functional traits at the assemblage level, factors such as interspecific interactions and the number of species that can be supported by primary productivity[24–26] must be considered at the assemblage level. These factors influence both the functional traits of individual species and the functional diversity of assemblages—critical properties for understanding ecological characteristics across different levels of biological organization. At the species level, thermal specialization refers to the breadth of a single species' thermal-tolerance range (TTrange = CTmax − CTmin); narrow ranges indicate specialists, while wider ranges indicate generalists. In contrast, assemblage-level diversity in thermal-adaptation strategies is quantified as the $n$-dimensional hypervolume formed by all species' CTmin–CTmax combinations. As such, an assemblage may simultaneously exhibit high specialization (many narrow-range species) and high hypervolume (a wide collective niche space) when specialists occupy distinct regions of trait space. However, how these three hypotheses and their underlying mechanisms affect thermal functional trait diversity at the assemblage level has not been directly studied within a comprehensive framework.

We address this knowledge gap by establishing eco-evolutionary simulation models based on these three hypotheses to generate testable predictions (Supplementary Table 1). We then use data from large-scale field experiments across latitudes and elevations to investigate how climate variability affects the functional traits of species and assemblages. We focus on moths, which, as small ectotherms, rely on thermal interactions with their environment to regulate body temperature, making them particularly vulnerable to temperature fluctuations[27,28]. Moths often exhibit remarkable species diversity in thermally stable tropical forests—for example, a single 16 km² Andean cloud-forest plot supports over 1100 geometrid species, more than 6% of global geometrid diversity[29]. The developmental rate of moths changes markedly with climate. Under warm conditions, many species complete their entire life-cycle from egg to adult in about 6–10 weeks, whereas at higher latitudes or in cooler environments, a generation can stretch to 3–4 months, with some species even requiring overwintering before eclosion. By contrast, the adult stage is typically very brief—most noctuid moths live only 7–14 days[30]. Accordingly, our model assumes that organismal lifespan exceeds short-term (e.g., daily) fluctuations but is shorter than long-term seasonal variation, enabling us to test how environmental variability across timescales

influences the evolution of thermal traits. Our integrated approach combines theoretical modeling with extensive empirical data to examine how environmental conditions shape thermal trait diversity at the assemblage level, providing insights into biodiversity responses to climate change.

## Results

### The eco-evolutionary models of favorability, short-term variability, and climatic variability hypotheses

We developed an eco-evolutionary and individual-based model to investigate how the mean and variability of temperature shape thermal traits at both species and assemblage scales. In this context, an assemblage refers to the complete set of individuals from all species that are currently alive and subjected to shared environmental conditions. A characteristic thermal performance curve is shared by all individuals of the same species, while the thermal trait diversity of an assemblage is characterized by the set of all viable species it contains.

In our framework, the mechanisms shaping thermal trait composition arise from two simultaneous processes: environmental filtering imposed by thermal regimes, and biotic interactions such as interspecific competition. The former determines which thermal strategies are viable, while the latter regulates the abundance and success of those strategies within assemblages (Fig. 1d). Abiotic filtering is imposed by thermal fluctuations, which determine reproduction and survival based on species' thermal performance curve. Biotic interaction occurs through competition, where the relative fitness of each species is defined as the product of its thermal performance and its population size. This formulation ensures that species that are both well-adapted to current conditions and numerically abundant have a greater influence on assemblage composition. For tractability, we initially constrain all species' thermal performance curves to have equal total performance area, establishing a trade-off between thermal specialization and generalization[31] (this constraint is later relaxed; Supplementary Fig. 1).

To test the favorability hypothesis, we manipulated mean environmental temperature and its effect on population dynamics. Higher environmental temperatures (up to 30 °C) increase resource availability and metabolic efficiency[11–14], leading to elevated population growth rates across species in the assemblage. Beyond this threshold, population growth rates decline due to physiological stress, eventually reaching zero. This relationship creates an unimodal productivity curve typical of biological systems (see Supplementary Fig. 2 for comparison with constant growth rate scenarios).

Temperature in our model is implemented through a hierarchical sampling process. At each time step, the actual temperature is drawn from a Gaussian distribution centered on a short-term average. This short-term average is itself updated every $T_{span}$ time steps by sampling from another Gaussian distribution centered on the mean ambient temperature. This dual-scale approach allows us to simulate both daily fluctuations and longer-term temperature patterns (detailed parameters in Supplementary Methods and Supplementary Table 2). We ran each simulation for $10^5$ time steps, with reproduction and mortality occurring in every step, to ensure stable species compositions were reached.

To analyze the model outcomes, we examined both the critical thermal limits composition and the niche-space hypervolume of the assemblages. The hypervolume analysis quantifies functional diversity by measuring the volume occupied by species in multi-dimensional trait space[32], using probability functions to describe trait distribution patterns while controlling for variation in species richness across assemblages (see "Methods" for details). This approach enabled us to assess how different thermal strategies distribute within assemblages and how functional trait space responds to environmental conditions. We specifically tested theoretical predictions for: (1) the favorability hypothesis by varying mean ambient temperature; (2) the short-term

variability hypothesis by altering short-term thermal fluctuations; and (3) the climatic variability hypothesis by modifying long-term thermal variability.

## Model predictions for the favorability, short-term variability, and climatic variability hypotheses

Our eco-evolutionary, individual-based model revealed that mean ambient temperature exerts the strongest influence on assemblage thermal traits. Increasing mean temperature led to a significant expansion in the assemblage's thermal trait space— quantified as the n-dimensional hypervolume encompassing species' thermal tolerances ($\beta = 1.01$, $p < 0.001$, $R^2 = 0.9$, 95% CI [0.79, 1.23]; Fig. 2a). Specifically, we observed that the upper bound of critical thermal minimum (CTmin) increased markedly with rising environmental temperature, while the lower bound of critical thermal maximum (CTmax) showed more modest changes. This pattern resulted in a steady increase in the assemblage's hypervolume as mean temperature rose (Fig. 2b), reflecting the influence of enhanced environmental productivity.

The model revealed greater thermal trait diversity at higher temperatures through two mechanisms. First, the range between CTmax and CTmin expanded. Second, thermal performance curves showed more diverse shapes, indicated by increased variance in thermal tolerance ranges (TTrange, a species-level attribute). This suggests that warmer environments can support species with a wider variety of thermal strategies. Notably, we observed an increase in thermal specialists—species with narrow thermal tolerance ranges—in high-temperature environments (Fig. 2c and Supplementary Fig. 3).

To characterize how warming reshapes assemblage composition, we used k-means clustering to classify species into four thermal strategies: warm-adapted (high CTmin), heat-tolerant (intermediate CTmin, high CTmax), cold-adapted (low CTmin), and heat-sensitive (intermediate CTmin, low CTmax). As mean temperature increased, warm-adapted and heat-tolerant species became more prevalent, while cold-adapted and heat-sensitive species declined (Fig. 3a, b). However, this diversification effect has limits—our model predicts that thermal trait diversity begins to decline when temperatures exceed 30 °C (Supplementary Fig. 4).

In contrast to mean temperature effects, short-term temperature fluctuations had surprisingly modest impacts on thermal trait composition. Neither the overall trait diversity (measured by hypervolume analysis) nor the variation in thermal tolerance breadth showed substantial changes with increasing daily temperature variability (Fig. 2d, e and Supplementary Fig. 5). The average thermal tolerance breadth of species showed only a slight decrease under greater short-term variability (Fig. 2f). While we observed minor shifts toward warm-adapted species and away from cold-adapted species, the relative proportions of thermal strategies remained largely stable (Fig. 3c, d).

Long-term climate variability produced distinct effects from short-term fluctuations. Increased seasonal temperature variation led to a modest reduction in overall trait diversity (Fig. 2g), driven by lower CTmin values and slightly higher CTmax upper bounds. This pattern resulted in larger average thermal tolerance ranges without increased variance in ranges, indicating selection for thermal generalist strategies (Fig. 2h, i and Supplementary Fig. 6). At the assemblage level, greater long-term variability reduced the proportion of warm-adapted specialists while slightly increasing cold-adapted and heat-sensitive species (Fig. 3e, f). We also found the results hold true with an alternative design of long-term variation (i.e., sine waves; Supplementary Fig. 7).

Importantly, our model revealed that, although high mean temperature and low long-term seasonal variation both appear to reduce selective pressures, they generate contrasting evolutionary outcomes. Favorability enables diverse strategies under high productivity, promoting the coexistence of both specialists and generalists. In contrast, low long-term seasonal variation consistently favors narrow-range

specialists by limiting the adaptive value of generalist strategies. These findings highlight how seemingly similar environments can produce divergent patterns of trait diversity. Our model demonstrates that mean temperature, rather than temperature variability, plays the dominant role in shaping thermal trait diversity. Warming promotes trait diversification primarily by enabling more specialist strategies through enhanced environmental productivity. While long-term climate variability selects for generalist strategies and slightly reduces overall trait diversity, short-term temperature fluctuations have minimal impact on assemblage thermal traits. These differential effects across temporal scales provide a mechanistic framework for predicting how climate change may reshape biodiversity patterns through both direct temperature effects and altered climate variability.

## Validating the generality of the models

In addition to generating predictions for the three main hypotheses, we extended our analyses for the eco-evolutionary model in multiple directions. First, we investigated the interaction between mean ambient temperature and short-term temperature variability on a broader scale of parameter combinations (Supplementary Fig. 8). We found that, invariably, increasing mean and short-term temperature variability both result in larger hypervolumes, higher TTranges, more surviving species, and longer average lifespans of species. Second, when we relaxed the constraint between the width and height of thermal performance curves, we found that the resulting effect on hypervolume, nonetheless, remains very consistent (Supplementary Fig. 1). Third, we tested a scenario where higher ambient temperatures no longer increase environmental productivity. Under this altered assumption, we observed no systematic trend in hypervolume across mean temperatures (Supplementary Fig. 2), confirming that the positive relationship between temperature and trait diversity in our main model arises through a productivity-mediated mechanism. In warmer environments, elevated growth rates reduce the extinction risk of rare species, thereby enhancing overall trait diversity.

To further explore the evolutionary mechanisms underlying trait diversity, we implemented an alternative scenario in which new species arise via mutation from resident species, simulating sympatric speciation. This alternative produced consistent results across all environmental scenarios (see Supplementary Fig. 9), which reinforces the generality of our conclusions. Finally, we reran the simulations with and without interspecific competition (see Supplementary Note and Supplementary Figs. 10–12). These comparisons revealed a two-step process: (i) environmental filtering imposed by the thermal regime, determining whether a species can persist, followed by (ii) interspecific competition, which further narrows the realized trait distribution through differential reproductive success. Together, these analyses underscore the robustness and mechanistic transparency of our framework.

## Empirical testing with moth assemblages

To test these theoretical predictions, we conducted field experiments measuring thermal tolerances of moths across three elevational gradients in Asia: Cameron Highlands, Malaysia (140–1959 m asl), Mt. Hehuan, Taiwan (343–3140 m asl), and Mt. Jiajin, Sichuan, China (860–4150 m asl). At each site, we established thermal testing stations at 500 m intervals to measure critical thermal maxima (CTmax) and minima (CTmin) under natural conditions (detailed methodology in Methods; experimental setup shown in Fig. 4a). Temperature loggers recorded ambient conditions throughout the experimental period.

Using our two complementary metrics—assemblage-level trait hypervolume (the n-dimensional space defined by CTmin–CTmax combinations) and species-level specialization (TTrange = CTmax−CTmin)—we found clear support for the favorability hypothesis. Hypervolume increased significantly with mean annual temperature

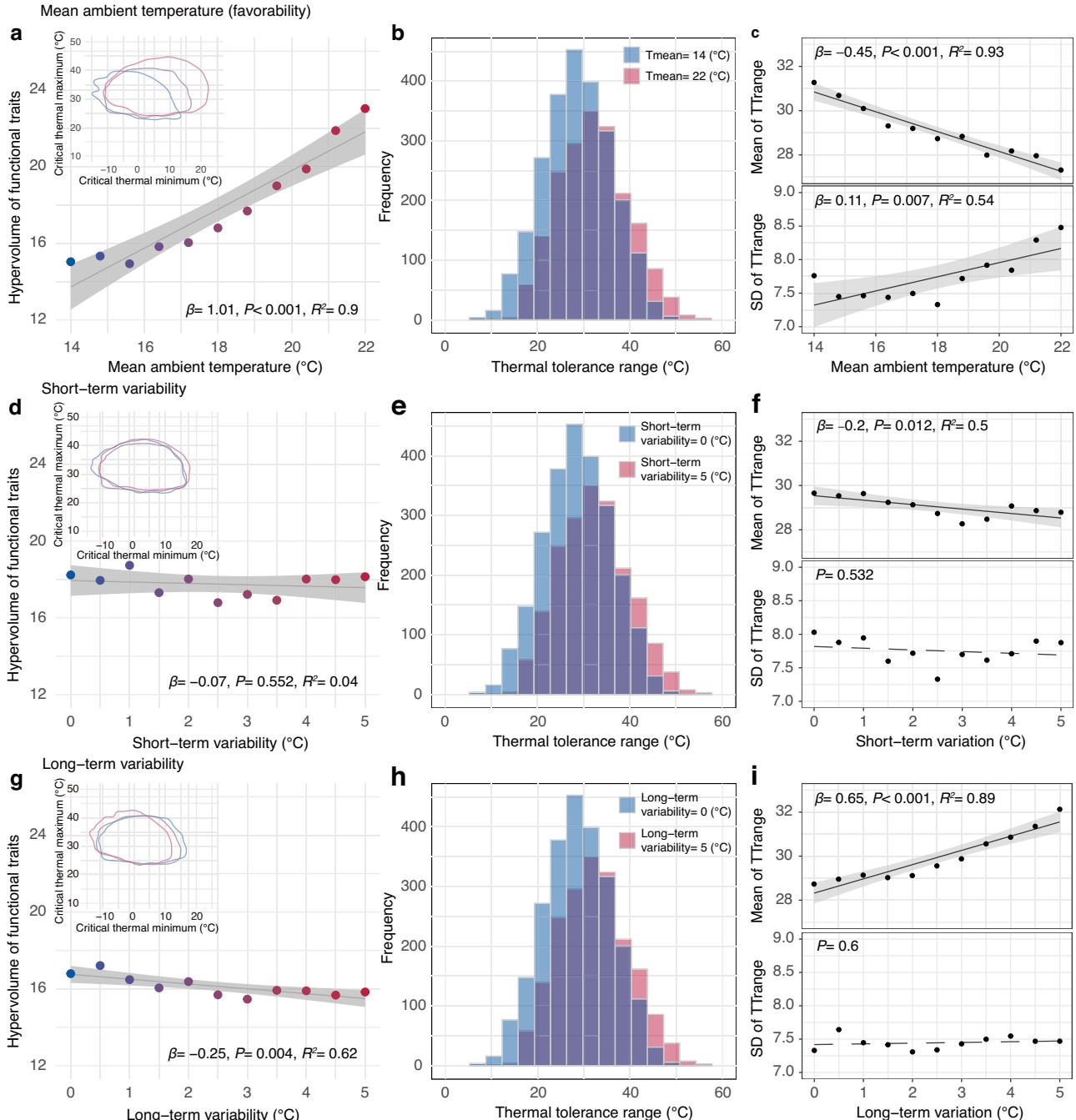

**Fig. 2 | Theoretical predictions of the impacts of the climatic properties on assemblage thermal traits. a** Effects of mean ambient temperature, showing a positive relationship with mean ambient temperature (Tmean) and carrying capacity of the environment, representing the favorability hypothesis. **b** Frequency distribution of thermal tolerance range (TTrange) comparing high (22 °C) and low (14 °C) Tmean. **c** Mean and standard deviation (SD) of TTrange in relation to mean ambient temperature. **d** Effects of short-term variability, defined as temperature variation that lasts for only a short period of time, represent the short-term variability hypothesis. **e** Frequency distribution of TTrange comparing high (5 °C) and low (0 °C) short-term variability. **f** Mean and SD of TTrange in relation to short-term variability. **g** Effects of long-term variability, defined as temperature variation that lasts for a long period of time, representing the climatic variability hypothesis. **h** Frequency distribution of TTrange comparing high (5 °C) and low (0 °C) long-term variability. **i** Mean and SD of TTrange in relation to long-term variability. Within **a**, **d**, **g**, the panel on the top-left corner shows the 95% quantile contours of critical thermal limits in three assemblages at low (1st, blue), middle (6tht, purple), and high (11tht, red) levels. In **a**, **c**, **d**, **f**, **g**, **i**, solid lines indicate significant relationships, dashed lines indicate insignificant relationships, and shaded areas represent the 95% confidence interval for the fitted regression line. Lastly, short-term variability is 2.5 and long-term is 0 in (**a**–**c**), Tmean is 18 and long-term variability is 2.5 in (**d**–**f**), Tmean is 18 and short-term variability is 2.5 in (**g**–**i**). For detailed results of linear regression models, see Supplementary Table 3.

(Tmean; $\beta = 0.7$, $p = 0.005$, $R^2 = 0.67$, 95% CI [0.29, 1.1]; Fig. 4b), indicating that warmer assemblages occupy a broader region of thermal niche space. At the same time, higher temperatures were linked to greater prevalence of thermal specialists, as shown by a decline in mean TTrange ($\beta = -0.2$, $p < 0.001$, $R^2 = 0.65$, 95% CI [−0.32, −0.08])

and a rise in its among-species variance ($\beta = 0.09$, $p = 0.013$, $R^2 = 0.46$, 95% CI [−0.04, 0.22]; Fig. 4c, d). Species-composition analyses corroborated this pattern, revealing significant increases in warm-adapted ($\beta = 0.02$, $p < 0.001$, $R^2 = 0.63$, 95% CI [0.01, 0.04]) and heat-tolerant species ($\beta = 0.06$, $p < 0.001$, $R^2 = 0.52$, 95% CI [0.04, 0.08]), alongside

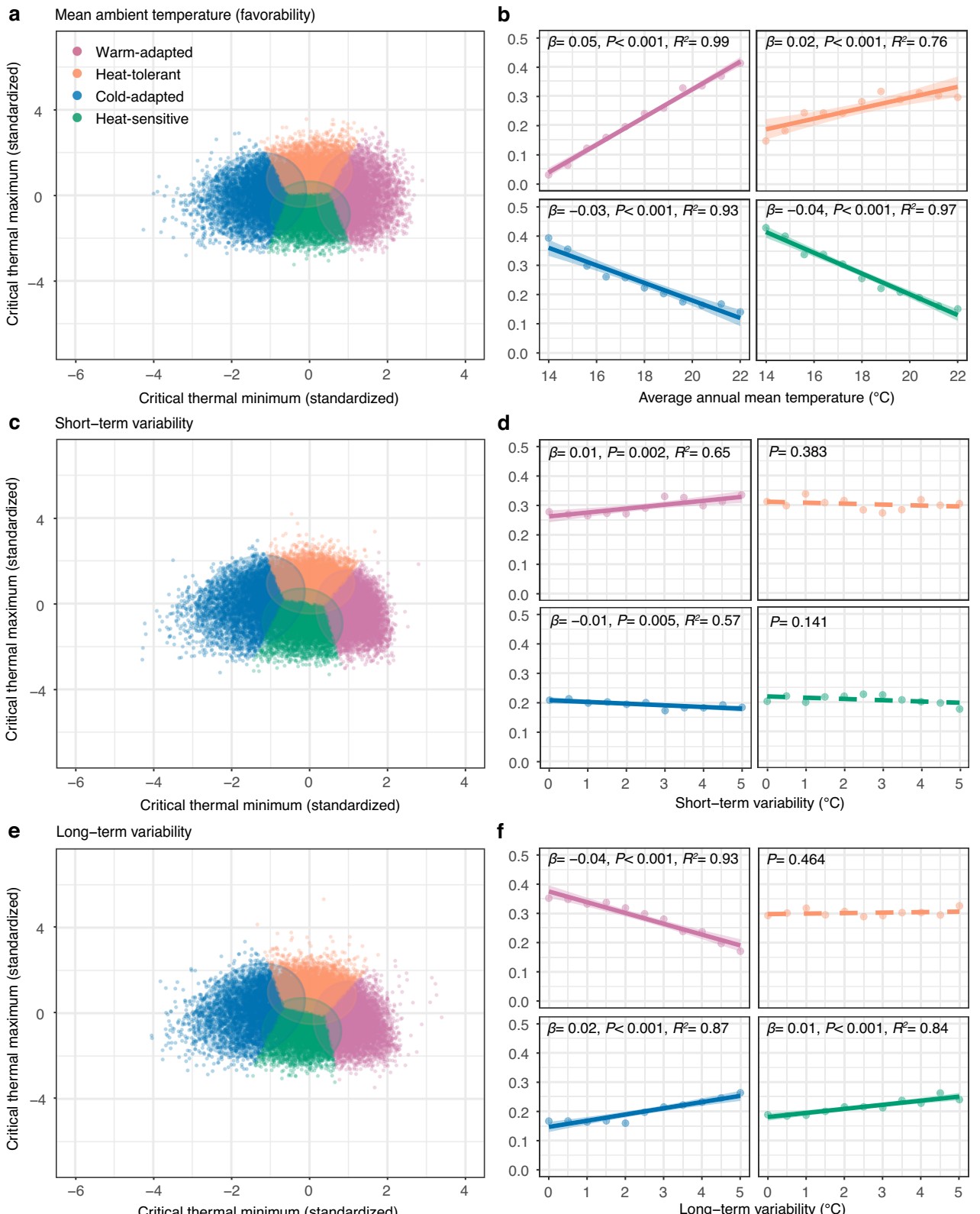

**Fig. 3 | K-means clusters distribution of critical thermal limits and changes of cluster ratio across different ambient gradients.** K-means clusters of model-simulated species (**a**) and ratio of 4-cluster species: warm-adapted species, heat-tolerant species, cold-adapted species and heat-sensitive species changes (**b**) across an average annual mean temperature gradient. K-means clusters of model-simulated species (**c**) and ratio of 4-cluster species changes (**d**) across the short-term variability gradient. K-means clusters of model-simulated species (**e**) and ratio of 4-cluster species changes (**f**) across long-term variability gradient. In **b**, **d**, **f**, solid lines indicate significant relationships, dashed lines indicate insignificant relationships, and shaded areas represent the 95% confidence interval for the fitted regression line. For detailed results of linear regression models, see Supplementary Table 4.

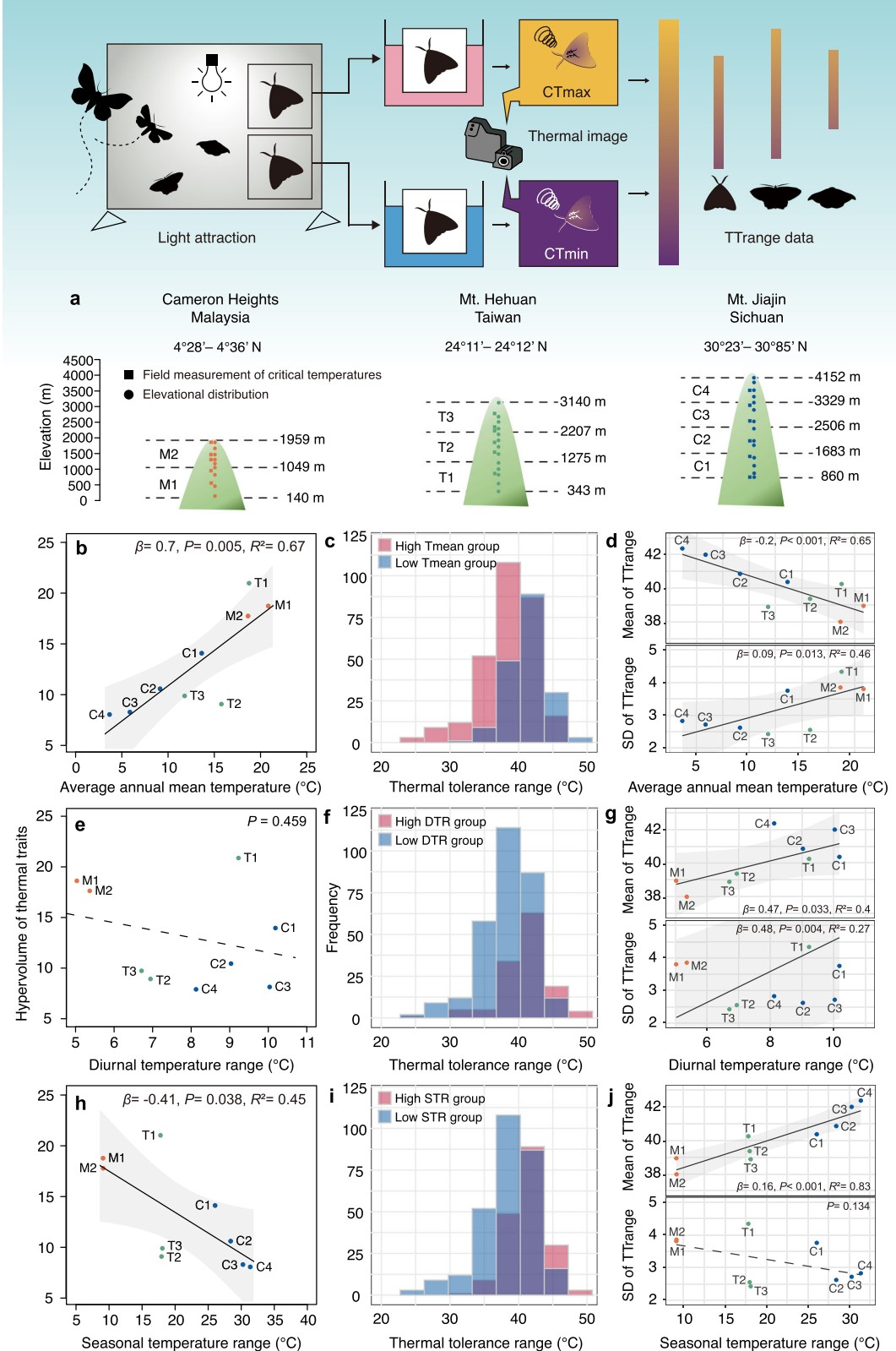

decreases in cold-adapted taxa ($\beta = -0.06$, $p < 0.001$, $R^2 = 0.69$, 95% CI [−0.09, −0.03]; Fig. 5b).

We used daily temperature range (DTR) to represent short-term variation, as it fluctuates more rapidly than seasonal variations and is an important environmental factor shaping species' adaptation to environmental variation[16]. We found DTR has no significant correlation with thermal trait hypervolume ($\beta = -0.74$, $P = 0.459$; Fig. 4e). However,

increased DTR was associated with more thermal generalists, as indicated by higher average TTrange ($\beta = 0.47$, $P = 0.033$, $R^2 = 0.4$, 95% CI [−0.54, 1.48]) and greater TTrange variation ($\beta = 0.48$, P = 0.004, $R^2 = 0.27$, 95% CI [0.07, 0.88]; Fig. 4f, g). Species composition remained stable across the DTR gradients (all $P > 0.05$; Fig. 5c). These results suggest limited effects of short-term environmental variation on assemblage thermal traits and agree with model predictions.

**Fig. 4 | Effects of mean temperature and climatic variability on thermal trait diversity at the assemblage level. a** Assemblages in each location. The methodology illustration is by Yun-Kae Kiang, created for this study without third-party content. **b** Relationship between average annual mean temperature (Tmean) and thermal trait hypervolume. **c** Histogram of frequency of thermal tolerance range (TTrange) for high (M2, T1, M1) and low (C4, C3, C2) Tmean groups. **d** Mean and standard deviation (SD) of TTrange as a function of Tmean. **e** Relationship between diurnal temperature range (DTR) and thermal trait hypervolume. **f** Histogram of frequency of TTrange for high (T1, C3, C1) and low (M1, M2, T3) DTR groups. **g** Mean and SD of TTrange as a function of DTR. **h** Relationship between seasonal temperature range (STR) and thermal trait hypervolume. **i** Histogram of frequency of TTrange for high (C2, C3, C4) and low (M2, M1, T1) STR groups. **j** Mean and SD of TTrange as a function of STR. In **b**, **d**, **e**, **g**, **h**, **j**, solid lines indicate significant relationships, dashed lines indicate insignificant relationships, and shaded areas represent the 95% confidence interval for the fitted regression line. For detailed results of linear regression models, see Supplementary Table 5. Underlying data files are provided in a permanent Zenodo repository under the accession code 17409650.

Following the literature convention, we use the seasonal temperature range (STR) to represent long-term thermal variations. Such analyses provided partial support for the climatic variability hypothesis. Thermal trait diversity decreased with increasing STR ($\beta = -0.41$, $p = 0.038$, $R^2 = 0.45$, 95% CI [−0.78, −0.03]; Fig. 4h). Higher STR promoted thermal generalists, shown by increased average TTrange ($\beta = 0.16$, $P < 0.001$, $R^2 = 0.83$, 95% CI [0.1, 0.22]; Fig. 4i, j). However, only cold-adapted species showed significant compositional changes with STR ($\beta = 0.02$, $P = 0.048$, $R^2 = 0.33$, 95% CI [−0.004, 0.04]; Fig. 5d), indicating limited predictive power for detailed assemblage composition. We further examined how environmental variables affect CTmax and CTmin (Supplementary Fig. 13). Using SHapley Additive exPlanations (SHAP) analysis (see Methods for details), we identified that CTmax is primarily driven by diurnal maximum temperature (DTmax), while CTmin is most strongly influenced by diurnal mean temperature (DTmean) (Supplementary Fig. 13a, b). These relationships were confirmed through linear regressions, showing significant positive associations between CTmax and DTmax ($\beta = 0.42$, $P < 0.001$, $R^2 = 0.3$, 95% CI [0.36, 0.46]; Supplementary Fig. 13g), and between CTmin and DTmean ($\beta = 0.44$, $P < 0.001$, $R^2 = 0.17$, 95% CI [0.32, 0.48]; Supplementary Fig. 13i).

Microclimate structure along the elevational gradients helps explain the assemblage-level patterns we report. First, our analyses controlled for potential sampling effects, as species richness did not significantly influence assemblage hypervolume ($P = 0.557$; Supplementary Fig. 14). These findings, derived from three distinct environmental gradients, provide robust evidence for the differential effects of mean temperature and temperature variability on thermal trait evolution, though future studies across additional taxa would help further improve the generality. Finally, to improve our understanding of the thermal conditions experienced by species across the elevation gradient, we plotted the relationships between annual mean temperature, STR and DTR against elevation (Supplementary Fig. 15). As expected, mean temperature decreased significantly with elevation, while STR increased and DTR remained relatively stable. We then examined microclimate heterogeneity (i.e., temperature variability) and found a significant three-way interaction among region, period, and elevation (LMM, Region × Period × Elevation, $P < 0.001$; Supplementary Fig. 16; Supplementary Table 7). Across all elevations and periods, Mt. Jiajin exhibited the highest temperature heterogeneity, followed by Cameron Highlands and then Mt. Hehuan ($P < 0.001$ for all post hoc comparisons; Supplementary Fig. 16; Supplementary Table 8). Daytime heterogeneity consistently exceeded nighttime heterogeneity in all regions ($P < 0.001$; Supplementary Fig. 16; Supplementary Table 9). Elevational trends differed among regions and diel periods. In Mt. Jiajin, temperature heterogeneity increased significantly with elevation during both day and night ($P < 0.001$ for both; Supplementary Fig. 16a; Supplementary Table 10). In Mt. Hehuan, daytime heterogeneity also increased with elevation ($P < 0.001$), whereas nighttime heterogeneity showed no significant trend ($P = 0.08$; Supplementary Fig. 16b; Supplementary Table 10). In Cameron Highlands, neither daytime nor nighttime heterogeneity exhibited significant elevational patterns ($P > 0.1$ for both; Supplementary Fig. 16c; Supplementary Table 10). Together, these macro- and microclimatic patterns provide an intuitive geographical context for interpreting the observed changes in thermal trait diversity.

## Discussion

Through an innovative combination of theoretical modeling and large-scale field experiments, our study reveals three key insights into how environmental factors influence thermal trait diversity in biological assemblages. First, warmer environments support more species and harbor greater diversity in thermal adaptations. These environments include assemblages of both narrow-range thermal specialists and broad-range generalists. Second, contrary to earlier theoretical expectations, short-term temperature variability exerts minimal influence on assemblage-level thermal trait composition. Third, long-term climatic variability fosters species with broader tolerances; however, its impact on overall trait composition is modest compared to the dominant effect of mean temperature.

Our eco-evolutionary model and empirical data support the favorability hypothesis, which states that benign, productive environments maintain a wider variety of thermal performance curves. This pattern emerges because reduced selective pressure allows both specialists and generalists to coexist. The resulting expansion of trait space in warmer assemblages aligns with classical theory linking high primary productivity to elevated species diversity[6,8,30], and extends that principle to functional-physiological diversity. Conversely, cooler assemblages are dominated by cold-tolerant yet heat-intolerant species, reflecting stronger environmental filtering that constrains trait hypervolume[9,18]. Our analysis further reveals distinct roles for short-term versus long-term temperature variability. Daily fluctuations surprisingly have little effect on critical limits or assemblage hypervolume, contradicting earlier two-species models whose simplifications evidently limit their predictive scope. On the other hand, while our data partially support Janzen's original climatic-variability hypothesis, the weak influence of long-term variability on assemblage composition suggests that its selective force is less pervasive than previously assumed.

Most sampled moths are nocturnal fliers that rely on shivering endothermy to elevate thoracic temperatures to approximately 33–38 °C[33], thereby buffering operative temperature fluctuations during activity. However, selective pressures on thermal physiology are also likely to act during periods of inactivity. Although adult moths typically seek shaded refugia during daylight hours, immobile life stages remain exposed to fine-scale microclimatic gradients shaped by topography and solar input. Microclimatic thermal heterogeneity exhibits significant three-way interactions among region, diel period, and elevation. Across multiple regions and time periods, high-elevation environments consistently show greater thermal heterogeneity[34], with daytime variability exceeding that of nighttime. Previous studies further indicate that in mountainous systems, heterogeneity is strongly shaped by topographic roughness, canopy architecture, and spatiotemporal variation in precipitation[34]. In the montane cloud forests of Taiwan, fog and low cloud layers exert powerful radiative control over the diurnal temperature range (DTR), producing nonlinear elevational gradients and marked discontinuities at mid- to high elevations[35]. These discontinuities align with zones of persistent fog and highlight the complex spatial structure of thermal

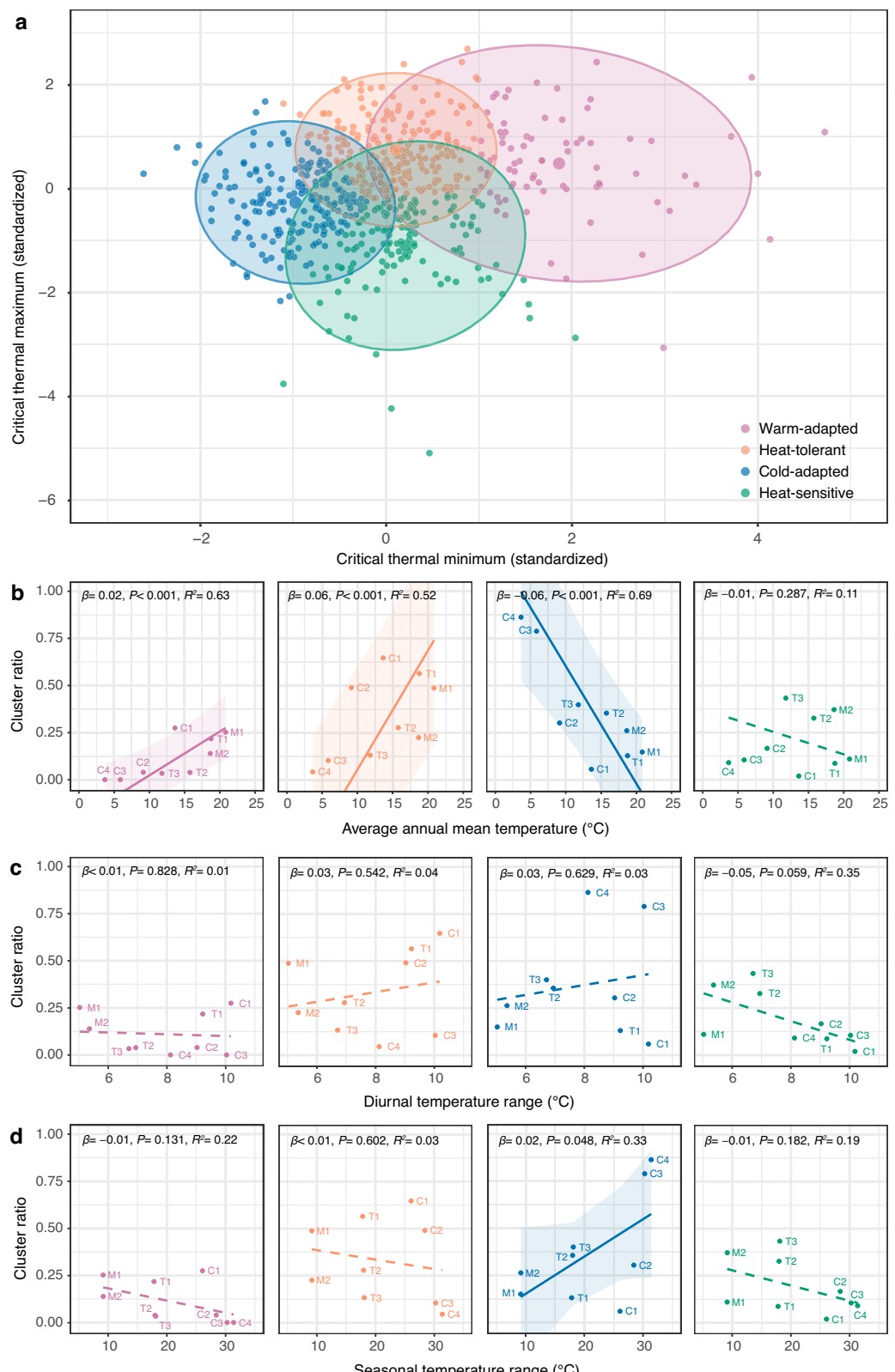

**Fig. 5 | K-means cluster distribution of critical thermal limits and changes of cluster ratio across mean temperature and climatic variability. a** K-means clusters of empirical species. Ratio of warm-adapted species, heat-tolerant species, cold-adapted species and heat-sensitive species changes across average annual mean temperature (**b**), diurnal temperature range (**c**) and seasonal temperature range (**d**). Solid lines indicate significant relationships, dashed lines indicate insignificant relationships, and shaded areas represent the 95% confidence interval of the regression line. For detailed results for the fitted regression line, see Supplementary Table 6. In **b**–**d**, texts next to the points represent different assemblages in Fig. 4a. Underlying data files are provided in a permanent Zenodo repository under the accession code 17409650.

regimes in montane ecosystems. Within this framework, environmental favorability—defined by the confluence of benign temperatures and high primary productivity—emerges as a principal driver of thermal trait diversity. Warm, productive environments support both thermal specialists and generalists, expanding trait hypervolume and increasing assemblage-level variability. By contrast, although high-elevation habitats exhibit substantial microclimatic variation, their overall thermal severity imposes strong environmental filtering that compresses trait distributions. Greater DTR may elevate CTmax in sedentary stages and increase average TTrange, yet this occurs without corresponding expansion in trait hypervolume. These findings support the idea that even short-term fluctuations, occurring on timescales shorter than species' generation lengths, can shape evolutionary outcomes distinct from those driven by long-term means. However, local environmental conditions—averaged over biologically relevant timescales—may exert stronger selective pressures than those reflected in long-term climatic averages alone[36].

Turning to longer time scales, the climatic variability hypothesis, assessed via STR, receives only partial support. Elevated STR broadens individual tolerance breadth and modestly increases the proportion of cold-adapted taxa; however, it contracts the overall hypervolume. In high-elevation forests, prolonged cool seasons impose strong selection for low CTmin, while summer extremes remain moderate. This keeps CTmax comparatively low and compresses trait space. Furthermore, multivoltine lowland species can adjust their phenology to align with favorable seasons. In contrast, many high-altitude congeners are univoltine and overwinter in diapause, which reduces their metabolic demand. These life-history constraints limit the spectrum of viable strategies where STR is large, yielding the narrower hypervolumes observed.

Collectively, these results align assemblage-level patterns with organismal biology and model expectations. Mean temperature, through its effects on productivity and microclimatic niche richness, emerges as the principal structuring axis of thermal diversity. Short-term variability has secondary effects within species, which are tempered by nocturnal ecology. Long-term seasonality defines lower tolerance bounds without broadening assemblage trait space. Understanding how different life stages and activity windows sample their thermal environments explains why favorability dominates in this moth system. This finding underscores the need to integrate behavior, phenology, and microclimate into future predictive frameworks.

Previous work on thermal tolerance has mainly focused on species-level responses[3,37,38]. These studies demonstrate that greater climatic variability often results in broader tolerance ranges, yet upper thermal limits do not necessarily increase toward the tropics. A critical distinction is that assemblage-level analyses examine mean values and variance across co-occurring species, while species-level studies typically track mean traits alone.

Taken together, our findings demonstrate how thermal trait distributions are shaped by the combined effects of environmental filtering and biotic interactions, which act as the core ecological and evolutionary mechanisms within assemblages. Our assemblage-scale approach reveals how thermal strategies are distributed within communities and clarifies which species occupy distinct functional positions, i.e., thermal niches. By integrating both immigration-based and local diversification scenarios, the model offers a flexible framework for understanding how thermal strategies evolve and persist under variable environmental conditions. To isolate environmental effects on competition, we held immigration/mutation rate constant across scenarios; although this can potentially overestimate adaptive responses under rapid change, it offers a clear baseline, and a natural next step is to relax this assumption to test the robustness of persistence predictions. While climatic variability can foster generalist species, its predictive power for assemblage composition remains limited, suggesting

that species- and assemblage-level perspectives provide complementary insights into the evolution of thermal traits.

Our findings provide a continental-scale investigation of assemblage-level thermal traits across multiple elevational gradients. The observation that warmer environments harbor greater thermal trait diversity emphasizes the conservation importance of low-elevation and low-latitude assemblages. However, it also raises urgent questions about vulnerability, as extreme warming events have already eroded functional diversity in other systems[14]. The heterogeneous effects of warming on different organisms underscore the need to identify which thermal trait combinations make species especially susceptible to climate change. Our theoretical framework provides a basis for such evaluations, although many questions remain regarding how thermal trait diversity influences ecosystem responses in the context of accelerating global change.

## Methods

### Field sampling design

The research took place in three mountain areas in East Asia, positioned at varying latitudes: Cameron Highlands, Malaysia (July 24–August 21, 2019; 4°28'0"–4°36'0"N, 101°11'0"–101°23'0"E), Mt. Hehuan, Taiwan (July 20–August 28, 2015; 24°16'0"–24°21'0"N, 121°10'0"–121°40'0"E), and Mt. Jiajin, Sichuan, China (June 28–July 22, 2017; 30°23'0"–30°51'0"N, 102°41'0"–102°54'0"E).

### Thermal tolerance measurements

Critical temperatures of species signify the cessation of primary ecological functions, like locomotion, assessed as the CTmax or minimum (CTmin) to assess how species adjust thermally to their environment[39]. Prior studies on the thermal tolerance of species were primarily lab-based[37,38,40,41]. However, the natural patterns of CTmin and CTmax, varying with elevation and latitude, likely stem from genetic factors paired with adaptive environmental responses[41]. Consequently, we captured individuals and tested their critical thermal tolerances directly in the field. This approach mirrors individuals' genuine physiological reactions to ambient temperatures. Historically, two thermal tolerance measurement methods were employed: static temperature and ramping temperature[42]. Different methodologies usually make cross-comparison of studies challenging. However, these differences have been reconciled recently using physiological models[42]. Therefore, in consideration of accessibility, we used the static temperature method for assessing moths' thermal tolerances.

We stationed thermal testing stations at roughly 500 m intervals along each elevation transect. Each station had a hot-water bath (WB212-B1, Double Eagle Enterprise Co., Ltd.; 10,398 cm³), a cold-water bath (BL720D, Yihder Technology Co., Ltd.; 19,440 cm³), and an infrared thermal camera (FLIR T420, FLIR Systems Inc., Danderyd, Sweden). The hot-water bath was set at 50 °C, controlled by a thermostat with a sensitivity range of 0 to 100 °C, while the cold-water bath was at −5 °C, controlled by a thermostat with a sensitivity range of −20–100 °C. The camera, with a 320 × 240-pixel resolution, captured temperature variances as slight as 0.045 °C, accurate within 2% of our testing range. To guarantee precision, the calibration service provided by FLIR Systems Inc. was applied.

We used a 100-watt lamp and a white screen to attract moths. We placed each moth in a sealed glass container (100*80*55 mm) for manipulation and observation. We confirmed that moth thorax temperatures matched the surroundings by the thermal camera. Thereafter, containers were immersed in either a cold- or a hot-water bath to observe the moth's reaction. We defined CTmin and CTmax as the temperature at which moths lost muscle control or showed spasms[43]. When individuals first showed signs of losing standing ability (i.e., inability to maintain an upright posture), we immediately removed the containers from the water bath and took thermal images of the

individual. Each moth was subjected to either CTmax or CTmin experiments, but not both[44,45].

After experiments, we used ThermaCAM Researcher Pro 2.10 for image analysis. Thermal images under the CTmax or CTmin experiments provided thorax temperature values of individuals[46,47]. The lowest and the highest temperature values (rounded to the first decimal place) in the thorax region were averaged as the critical temperature for that individual. Average CTmax and CTmin for species were derived from individual records. In total, we assessed 1475 individuals in Malaysia, 2257 in Taiwan, and 1917 in China. We deposited all specimens at the Biodiversity Research Museum, Academia Sinica, Taipei, Taiwan.

## Identification of moth species
We initially identified specimens to the morpho-species level based on their morphological characteristics, with each being assigned a unique name code. Subsequently, we consulted relevant academic literature, especially the faunistic reviews in sampling sites or neighboring regions[48–50], to determine the formal scientific names of these morphospecies. Once the scientific names were confirmed, we used these names to record the specimens. When visual differences between taxa were vague or when polymorphisms were hard to identify, we dissected the genitalia to facilitate recognition. We included species that exhibited thermal attributes (both CTmax and CTmin) for our study. Altogether, we found 16 families with 264 species in Malaysia, 14 families with 157 species in Taiwan, and 15 families with 232 species in China.

## Environmental data collection
We positioned iButton thermometers (Maxim Integrated Products, Inc.) at 250 m elevation intervals near each moth light trap site, housing each device in a T-shaped PVC tube mounted 1.5 m above ground level. This setup, following standard meteorological protocols, protected sensors from direct solar radiation and rain while ensuring adequate ventilation. We validated measurement accuracy by calibrating iButton readings against a standard weather station at the Department of Atmospheric Sciences, National Taiwan University. Temperature recordings were taken every 30 min throughout the experimental periods (July 24–August 21, 2019, in Cameron Highlands; July 20–August 28, 2015, in Mt. Hehuan; June 28–July 22, 2017, in Mt. Jiajin). For each site, we calculated the daily mean temperature from the recording periods and derived the DTR from the difference between the daily maximum and minimum temperatures. We assigned ambient temperature to each species based on the midpoint of its elevational distribution, using linear interpolation between the two nearest elevation temperatures when direct measurements were unavailable.

For seasonal temperature, we sourced monthly temperature records from CHELSA (version 2.1)[51] spanning 1990 to 2019. We followed the ANUCLIM criteria when determining the annual temperature[52]. The average annual mean temperature is the average of the mean daily air temperature for each month from 1990 to 2019. The STR is the difference between the average maximum temperature of the warmest month and the average minimum temperature of the coldest month, where the former corresponds to the mean of July mean daily maximum air temperature across the 30-year span, and the latter corresponds to the mean of January mean daily minimum air temperature over the same 30-year duration.

## Thermal trait hypervolume of assemblages
We used the "hypervolume_gaussian" function from R package "hypervolume"[32] (v3.1.6) in R (v4.3.0) to calculate the $n$-dimensional kernel density of each assemblage to represent thermal trait diversity. The hypervolume method can make the measurement of biodiversity independent of the number of species, because it focuses on the distribution and coverage of functional traits in the multidimensional space. To further avoid the effects of the number of species, we randomly sampled 20 species when calculating the hypervolumes (and 100 species for mean ambient temperature, diurnal and seasonal temperature range group analyses), repeated 100 times, and used the averaged value to summarize each assemblage. To avoid the influence of the scale difference of traits, we used standardized CTmax and CTmin of species as the thermal traits and used the ratio of sample size for each species to weight the random points in the hypervolume for each assemblage. Finally, we used linear mixed-effects models (R package "lme4"[53] v1.1.36) to examine the linear relationship between the hypervolume of assemblages and ambient temperature measurements.

## Kernel density estimation and $k$-means clusters
We used the R package "ks"[54] (v1.14.3) for two-dimensional kernel density estimation[55] to estimate the density and position of a given variable in the trait space defined by CTmax and CTmin. The optimal bandwidth of the smoothing kernel was determined by the "samse" pilot bandwidth selector[56]. To characterize and compare thermal trait compositions between theoretical predictions and empirical assemblages, we employed k-means clustering (R base function "kmeans" v4.3.0). Sensitivity analyses with different k values ($k = 2–4$) revealed that $k = 4$ provided the most biologically interpretable groupings, corresponding to four distinct thermal strategies: warm-adapted species (high CTmin), heat-tolerant species (intermediate CTmin with high CTmax), cold-adapted species (low CTmin), and heat-sensitive species (intermediate CTmin with low CTmax). This clustering scheme facilitated meaningful interpretation of thermal strategies while maintaining robustness, as the concordance between theoretical and empirical patterns remained consistent across different clustering approaches.

## SHapley additive exPlanations (SHAP) analysis
Analyses were run in Python v3.11.5. To assess the relative importance of environmental variables in predicting critical thermal limits (CTmax and CTmin), we applied SHAP analysis using "TreeExplainer" function (from package "shap" v0.42.1)[57,58] to random-forest regression models fitted with "RandomForestRegressor" (from package "scikit-learn" v1.3.0)[59]. SHAP values quantify the marginal contribution of each predictor to model outputs by averaging over all possible orderings of feature inclusion. This method allows for interpretable, model-agnostic estimation of variable importance, while accounting for interactions among predictors. We report the mean absolute SHAP value for each environmental variable, which reflects its overall influence on predicted thermal limits.

## Elevational patterns and diurnal variation in forest microclimate heterogeneity across three biogeographic regions
To assess variation in forest microclimate heterogeneity along elevational gradients across three geographic regions—Cameron Highlands, Malaysia; Mt. Hehuan, Taiwan; and Mt. Jiajin, Sichuan, China—and to compare diurnal and nocturnal patterns, we derived gridded hourly macroclimate data for July (the warmest month during the field sampling years) from the ERA5 reanalysis dataset[60] using the "mcera5" R package[61] (v0.4.0). Microclimatic conditions for each sampling point were then modeled using the "microclima" package[62] (v0.1.0) in conjunction with "NicheMapR"[63].

For each sampling site, a 500-m buffer was established, within which four random points were selected in addition to the central sampling point ($n = 5$ per site). Digital elevation models (30-m resolution) were retrieved for each point using the "get_dem" function in "microclima". Concurrently, 30-m resolution habitat classification data were obtained from the GLC_FCS30D dataset[64], providing forest type inputs essential for microclimate modeling. Although GLC_FCS30D

distinguishes between open and closed canopy types, this differentiation was ignored during modeling (e.g., both "Open" and "Closed Evergreen Broadleaved Forest" were categorized as habitat = "Evergreen Broadleaf forest" in "microclima"). Only forested points were retained for subsequent analyses.

Hourly temperature estimates were computed for five vertical strata−forest floor (0.05 m), top of the herbaceous layer (0.25 m), low-shrub canopy (0.5 m), breast height (1.5 m), and sub-canopy (2.5 m)−using the "runauto" function in "microclima" over the sampling interval (July 12–18) of the respective years. Notably, breast height (1.5 m) represents a standard level for understory light measurements[65,66], while the other strata may differentially influence moth activity across life stages.

Day and night periods were delineated based on local sunrise and sunset times: 23:00–12:00 UTC for Mt. Jiajin, 00:00–11:00 UTC for Cameron Highlands, and 22:00–10:00 UTC for Mt. Hehuan. Microclimate heterogeneity was quantified as the standard deviation of temperature across the five vertical strata for each hour, separately for daytime and nighttime. Daily values were averaged across the 7-day sampling window.

We fitted linear mixed models using the "lme4" R package[53], treating microclimate heterogeneity as the response variable, with geographic region, diel period (day vs. night), and elevation as fixed effects, and sampling point ID as a random effect. Where significant interactions were detected, post hoc pairwise comparisons (Tukey tests) and slope comparisons (using the "emtrends" function with Kenward–Roger correction) were conducted with the "emmeans" package[67] (v1.8.8).

Further methodological details, including the eco-evolutionary model framework and simulation procedures, are provided in the Supplementary Methods.

## Reporting summary
Further information on research design is available in the Nature Portfolio Reporting Summary linked to this article.

## Data availability
All data generated or used in this study have been deposited in the Zenodo repository under the accession code 17409650[68].

## Code availability
The analysis code used in this study has been deposited in the Zenodo repository under the accession code 17409650[68].

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

## Acknowledgements

The authors would like to acknowledge the use of AI-assisted technologies in the preparation of this manuscript. ChatGPT and Claude were used to assist with text editing and refinement. The use of AI assistance was limited to text editing and did not extend to data analysis, interpretation, or drawing of conclusions. We also want to thank Dr. Robert K. Colwell and Dr. Jennifer Sunday for helping us improve an earlier version of the manuscript. The authors are grateful for the computational resource supported by Stuart West and the European Research Council (Horizon 2020 Advanced Grant 834164). Academia Sinica grant AS-SS-106-05 (SFS, ICC). Academia Sinica grant AS-SS-110-05 (SFS, ICC). Ministry of Science and Technology of Taiwan grant 100-2621-B-001-004-MY3 (SFS). Ministry of Science and Technology of Taiwan grant 104-2311-B-001-028-MY3 (SFS). Ministry of Science and Technology of Taiwan grant 108-2314-B-001-009-MY3 (SFS). Ministry of Science and Technology of Taiwan grant 104-2311-B-006-006-MY3 (ICC).

## Author contributions

S.-F.S. conceptualized the study. T.-M.H., S.W., Mark L., G.-S.M., Y.-S.J., C.-C.H., C.-Y.H., C.-H.W., M.-N.T., S.-F.C., I.-C.C. and S.-F.S. developed the methodology. Mark L., T.-M.H., S.W., G.-S.M., Y.-S.J., C.-C.H., C.-Y.H., C.-H.W., M.-N.T., I.-C.C. and S.-F.S. conducted the investigation. Ming L. and S.-F.S. built and analyzed the models. T.-M.H. and Ming L. generated

the visualizations. S.-F.S. and I.-C.C. acquired funding and supervised the work. S.-F.S., Ming L. and T.-M.H. wrote the original draft. Ming L., T.-M.H., S.W., I.-C.C. and S.-F.S. reviewed and edited the manuscript.

## Competing interests

The authors declare no competing interests.
