## [Transparent Peer Review file · Nature Communications]

Warmer environments harbor greater thermal trait diversity in moth assemblages

Corresponding Author: Dr Sheng-Feng Shen

Version 0:

Reviewer comments:

Reviewer #1

(Remarks to the Author)

This paper provides a highly original combination of modeling and data to test a core set of hypotheses about geographic patterns in thermal traits of insects. The authors set up three main hypotheses about drivers of assemblage-level diversity (favorability hypotheses, the climate variability hypothesis, and the short-term variability hypothesis). To test the hypotheses, they first develop an eco-evolutionary model that distributes 'species' with sampled traits across environments, impose several types of environmental variation on them, track mortality and reproduction, and then ask what the characteristics are of the assemblages that make it through environmental filtering. The main result is that environments with higher mean temperatures contain (post-filter) a greater diversity of thermal performance types (supporting the favorability hypothesis). They then test these model predictions with truly epic data collection – on three independent altitudinal gradients in Asia (gradients of 1800 – 3200 m), in which they captured well over 5000 individual moths and measured either CT_{min} or CT_{max} on each one, then identified to species. The data also appear to support the favorability hypothesis, in that there was a greater diversity of tolerance limits in locations with higher annual mean temperatures.

This paper is thought-provoking, and I agree with the authors that it's one of the first (if not the first) to do such an extensive assemblage-level analysis. This will be of broad interest to the large group of thermal biologists in the world.

That said, I have two broad conceptual points about the approach, analyses, and conclusions, then a third point about raw versus derived data.

1. The idea of using environmental temperature (averaged over whatever spatial or temporal scales) as a proxy for what moths experience is fraught with nuance. For example, many but not all moths raise their body (at least thoracic) temperatures well above ambient air temperatures when they fly. The moths in this study all were flying when they were captured (at light traps) at night, and undoubtedly many of them had thoracic temperatures 5 – 10 degrees or more above ambient air temperatures. For older examples in the literature, see work on hawkmoths by Bernd Heinrich, who shows that thoracic temperatures can be regulated to around 35 C almost regardless of ambient air temperatures.

If moths raise their thoracic temperatures to more or less the same level for flight (don't know if this is actually the case, but it seems like a good starting – null – hypothesis), then this would suggest that species living predominately in lowlands, in higher mean annual temperatures, could evolve small thermal tolerance ranges (TTrange) because (i) they regularly experience [during flight] thoracic temperatures above ambient temperatures while (ii) at the same time they rarely experience very low temperatures, such that there is little selective pressure for them to keep physiological integrity in the cold. By contrast, species living in colder highlands would (i) experience the same high body temperatures in order to fly (even in the cold) but would also experience much colder body temperatures at rest/night/seasonally and thus may evolve lower CT_{min}. If I'm right about this, then these patterns of exposure to temperature during activity and rest could drive patterns much like those seen in 4d. There's perhaps a separate problem with understanding elevational patterns of SD but I think similar thinking here could help elucidate this.

Follow up points from the above. (A) Perhaps the mechanisms I describe above fit already into the schemes outlined by the authors, but none of this is made explicit. There needs to be stronger connection to the physiology and body temperatures of moths in their environments. (B) I was surprised to see no discussion of diurnal patterns of activity and flight, given that most

(but not all) moths are night-flying.

2. My second point is related to the first in the sense that it's another process that can decouple drivers of moth body temperatures from 'ambient temperatures.' I'm referring here to microclimates, which have received a lot of attention recently in the literature on insects (and other small ectotherms). The idea is that coarse measures of air temperature in space and time can do a poor job of capturing thermal experiences of actual insects, which tend to be much more diverse. I.e., there is typically a lot of thermal diversity across space and time that is not accounted for by data loggers and mean air temperatures. It's possible that this problem would be most acute for animals operating in the daytime, during which differences in sun and shade can drive large differences in operative temperatures at very fine spatial scales. This may be less of a concern for night-active animals like moths, as nighttime microclimatic temperatures probably are less diverse. Nevertheless, moths live also through daytimes, and likely experience quite substantial variation in daytime temperatures experienced – and this variation may differ according to elevation or species. For example, if lowlands are sunnier on average than highlands, microclimatic diversity in operative temperatures may be on average higher in lowlands. If individual species use this diversity on average in different ways, then long-term evolution of assemblages could lead to patterns like those seen in Figure 4d. Here too, maybe this is just a more mechanistic description of what the authors are suggesting already, but it's not clear to me that this is the case – plus I was surprised to see nothing about microclimates mentioned anywhere in the manuscript.

3. My other complaint is about how the data and analyses are presented. Overall, the approach is to present quite abstract and derived version of the data (both modeled and empirical) – things like hypervolumes and K-mean cluster distributions. In addition, the basic data on temperature tolerances are rendered as TTRange (thermal tolerance range) and the standard deviation of TTRange. These latter two are more approachable, but they're still derived. For example, I wondered whether CTmax or CTmin changes more with the various temperature measurements. I think I can figure that out but nowhere are there plots of each of those things individually against the independent variables.

Likewise for temperature data – like it would be nice to have max and min temperatures for the different sites plotted separately, perhaps against altitude.

I understand that the current approach is more compact in that it summarizes lots of information within single parameters, but the cost is that it's more difficult to understand what's happening within the raw data to drive those changes.

Line 233: technically the measurements of wild moths do not constitute 'thermal performance curves' but rather tolerance ranges or critical thermal limits.

(Remarks on code availability)

Reviewer #2

(Remarks to the Author)

The article combines modelling and field data from moth species in Asia, across an elevation gradient, to test predictions derived from three selected hypotheses. The hypotheses aim at explaining biodiversity patterns in thermal adaptations, and were named: favorability (F), long-term seasonal variation (LT), and short-term daily fluctuations (ST). The results support F in the context of a tropical region, which, citing the article words (line 235), "aligns with classical theory". Thus, I would recommend to clearly frame the novelty of the study, beyond referencing the scale of the study (line 257). However, I have concerns regarding (i) conceptual foundations and the writing of the manuscript, and (ii) model communication and implementation. Below I summarize my concerns and provide suggestions for improvements. However, the list is not exhaustive as my decision is to not recommend this article for publication in Nature Communications journal.

(i) conceptual foundations and the writing of the manuscript

The writing feels rush. However, papers are messages and I would recommend to take the time to frame, write and deliver the messages such that they contribute to progress in the way intended by the authors. To this work, the modelling part is as important as the field work. However, model description could benefit from appropriate justification of modelling decisions, and from explaining the terminology and acronyms being used (section (ii) expands on this issue).

The article says that the cited hypotheses are competing hypotheses. However, this is not clear to me. Actually, observations on patterns of thermal adaptations of species could rise due to more than to single reasons/hypotheses. Eg., F and LT could get closer to what is expected from "constant" environments, which would lead to reduced plasticity and higher specialization (I recommend to check the literature on phenotypic plasticity theory, many are based on individual-based models). Further, if the life cycle of the species in question is longer than the environmental fluctuations (eg., under ST, which is the case for the moths), the mean environment could strongly lead the eco-evolutionary dynamics. I've seen that the study has come across this in lines 228-230.

In the abstract and the main text, there is mentioning and discussion on the mechanisms shaping thermal adaptation. However, there is little if any, mention to evolutionary mechanisms. Thus, what is meant by mechanisms in the context of this study? I expect that understanding the mechanisms means understanding the why the patterns we observe, which would be very interesting. However, I do not see that here.

lines 5-7 (and elsewhere): what is the difference between thermal specialization and great diversity in thermal adaptation strategies? For instance, an assemblage of species with a high diversity of thermal adaptation strategies can result from being composed out of many specialist...

line 26: emergent properties are a characteristic of the agent-based (also called individual-based) approach. Therefore, species level studies can potentially display emergent properties as well, depending on how they are modeled.

line 57: why such distinction represents an advance to our understanding? Please clarify

line 65: "has never" Drop this, and rather focus on the uniqueness and potential of the study

lines 71-75: include here an statement of what is known about the diversity of moths. Are they more diverse in lower variable - tropical - environments than in "highly" fluctuating temperate ones? Also, information on the life cycle is needed. To my knowledge moth's life cycle is days or months. Not a single day. How does this fact affect the modelling of the daily vs long-term fluctuations?

lines 122-124: to what extent could this result from model assumptions? ie., soft vs abrupt decays in thermal tolerance below vs above optimal temperature (S4), plus the imposed upper temperature maximum limit (S1)

lines 127-132: Can one discard that there are not two extreme strategies in the assemblage that result in high variance. Thus, no diversity in strategies. Just high variation between few strategies.

(ii) Model communication and implementation

The main text will benefit very much from having precise information and justification regarding model selection, assumptions, decisions, and explanations - e.g., regarding model terminology, crucial: what is an assemblage to the model?. The supporting information needs considerable work with the order, sequence and drafting in general. I would suggest to find a good modelling paper to use as a reference. Particularly, when deciding for parameter names and subscripts. At the moment is all very confusing. As I mentioned above, the modelling is an important step in this study, and deserves to be conveyed as such.

Supplementary text

The eco-evolutionary model

- The section "overview" needs to start with a clear motivation and objectives of the modelling.

Also, model selection and decisions need to be justified and model concepts, traits and measurements, clearly explained. E.g., it is not known what an assemblage is (to the model) until the very end (and still its definition is not precise). This needs to be at the beginning. Is the model spatially explicit or implicit? what are the basic entities considered "individuals" in the model? the species?

how do "individuals" differ? in what traits?

- Clearly define what is considered an assemblage in the context of the model. It could be fair (to the reader) to assume that each patch is an assemblage of species; while later an assemblage is barely defined as some consideration of all species across all patches... This definition needs to be crystal clear.

- Furthermore, it feels that evolution is lacking even given that it is said to be an eco-evolutionary model. To me it seems that the model is purely ecological with emigration/immigration dynamics. As it is at the moment, I would suggest to drop "eco-evolutionary" from the name.

- Then, why to assume explicit space, ie., many patches, instead of implicit space with N_t species defining the assemblage? this is not explained. Wouldn't that be simpler? Please justify

- Why to use migrations (potentially increasing species number) and not mutations, which leads to variations of existing species? Please justify. Further, how migration rate is adjusted to fit the different time scales? for example, considering daily fluctuations, moths immigrants per day would be expected to be much lower than moth immigrants per days or months.

- I would recommend to use appropriate functions to model the fluctuations in T such that the expected behavior per hypothesis is achieved by changing one or few parameter(s) of the same function. Eg., if going after a stochastic model: why not to use white noise vs red noise auto-correlations? e.g., daily temperature fluctuations can be modeled using white noise auto-correlations, while long-term, using red-noise (positive) auto-correlations. This will involve taking the current temperature value in only one step, and the two scenarios could be implemented by changing the value of one parameter. Alternatively, why not to use a deterministic model? E.g., a sine function with different period T depending on the environmental scenario. Wouldn't a deterministic model help making more precise predictions (ie., reduce the effect of stochasticity).

Anyway, the type of selected model needs to be justified.

- There are model decisions taken to ensure "fair competition". What is that? This phrase does not seem biological.

- In S5, "min" is written as a function, not as an argument. Please clarify. Furthermore, what is a coefficient for weighting? Also, please explain how the min function works. It seems to take the minimum value from a vector that contains the

coefficient for weighting, the density of species j at patch i , and the unity? Thus, p_{compete} can be less or equal to 0.5. However, this is not clear.

- regarding mortality, it is not clear whether it means that a species either goes extinct or not with given probability; or an individual inside a species dies with the given probability.

- "Because allopatric speciation is much easier to evolve than sympatric speciation". Given the current level of knowledge, allopatric speciation is easier to understand than sympatric speciation. This does not mean that it is much easier to evolve. If someone has demonstrated such a statement then add the citation here.

- Again, it is not clear what an assemblage is. Is it the result of averaging, summing up, or something else, all the patches? If it consists of averaging, then letting offspring migrate might have an impact on the resulting assemblage. Immigration is already implemented by arrivals of random species from the outside. On the other hand, how the variation was changed for the variability hypotheses? this is not clear, and this is crucial for the resulting predictions. Besides, immigration and reproduction dynamics might also differ depending on the variability hypothesis in question. E.g., in the case of daily fluctuations, do moth species undergo reproduction each day? At the moment, modelling decisions seem to suggest that, which would not match the real case study.

(Remarks on code availability)

Version 1:

Reviewer comments:

Reviewer #1

(Remarks to the Author)

Thanks for the thorough set of responses and edits made in response to the previous round of reviews. The manuscript appears significantly stronger for it, and I'm largely satisfied with the new draft.

The one area that still needs attention is the Discussion. In response to my prior comments, you added sections on shivering endothermy and on microclimates. This is good overall, but the writing is odd here. There are two paragraphs that both start with almost identical topic sentences "Most sampled moths are nocturnal fliers that elevate thoracic temperatures to ~33–38 °C via shivering endothermy." Some of the other information in the paragraph also is redundant. I would suggest either (1) combining the paragraph into one, so that you're focusing on two related issues that may alter the thermal experience of moths (although I can see that this would be a giant paragraph), or (2) structuring the paragraphs so that they're more distinct from the beginning, one on flight- or shivering-generated heat production and the other on microclimatic choices. These really are different, as one comes from within the moths and the other from the outside, based on the choices they make about where to go and when to be active.

(Remarks on code availability)

Reviewer #2

(Remarks to the Author)

The authors did a good job addressing my concerns and improving the communication of the model.

I have a minor comment:

- On the use of fixed immigration / mutation rates across environmental scenarios: One needs to be cautious with this assumption, since it can lead to an overestimation of the adaptive response of populations. This is particularly relevant under scenarios of "fast" (relative to generation time) change. Thus, for instance the model could predict persistence when in reality extinction is observed. Though to the purpose of the study such model assumption could be justified, please add a remark to make readers aware of this.

(Remarks on code availability)

The code is written in C language. It has a read-me file with directions to the source and data files. Potential future users will need to find their way by navigating and experiencing the code. Though the source code is commented, it might not be accessible for users without knowledge on C programming language.

Reviewer's comments 1:

REVIEWER COMMENTS

Reviewer #1 (Remarks to the Author):

This paper provides a highly original combination of modeling and data to test a core set of hypotheses about geographic patterns in thermal traits of insects. The authors set up three main hypotheses about drivers of assemblage-level diversity (favorability hypotheses, the climate variability hypothesis, and the short-term variability hypothesis). To test the hypotheses, they first develop an eco-evolutionary model that distributes 'species' with sampled traits across environments, impose several types of environmental variation on them, track mortality and reproduction, and then ask what the characteristics are of the assemblages that make it through environmental filtering. The main result is that environments with higher mean temperatures contain (post-filter) a greater diversity of thermal performance types (supporting the favorability hypothesis). They then test these model predictions with truly epic data collection – on three independent altitudinal gradients in Asia (gradients of 1800 – 3200 m), in which they captured well over 5000 individual moths and measured either CTmin or CTmax on each one, then identified to species. The data also appear to support the favorability hypothesis, in that there was a greater diversity of tolerance limits in locations with higher annual mean temperatures.

This paper is thought-provoking, and I agree with the authors that it's one of the first (if not the first) to do such an extensive assemblage-level analysis. This will be of broad interest to the large group of thermal biologists in the world.

That said, I have two broad conceptual points about the approach, analyses, and conclusions, then a third point about raw versus derived data.

1. The idea of using environmental temperature (averaged over whatever spatial or temporal scales) as a proxy for what moths experience is fraught with nuance. For example, many but not all moths raise their body (at least thoracic) temperatures well above ambient air temperatures when they fly. The moths in this study all were flying when they were captured (at light traps)

at night, and undoubtedly many of them had thoracic temperatures 5 – 10 degrees or more above ambient air temperatures. For older examples in the literature, see work on hawkmoths by Bernd Heinrich, who shows that thoracic temperatures can be regulated to around 35 C almost regardless of ambient air temperatures.

If moths raise their thoracic temperatures to more or less the same level for flight (don't know if this is actually the case, but it seems like a good starting – null – hypothesis), then this would suggest that species living predominately in lowlands, in higher mean annual temperatures, could evolve small thermal tolerance ranges (TTrange) because (i) they regularly experience [during flight] thoracic temperatures above ambient temperatures while (ii) at the same time they rarely experience very low temperatures, such that there is little selective pressure for them to keep physiological integrity in the cold. By contrast, species living in colder highlands would (i) experience the same high body temperatures in order to fly (even in the cold) but would also experience much colder body temperatures at rest/night/seasonally and thus may evolve lower CTmin. If I'm right about this, then these patterns of exposure to temperature during activity and rest could drive patterns much like those seen in 4d. There's perhaps a separate problem with understanding elevational patterns of SD but I think similar thinking here could help elucidate this. Follow up points from the above. (A) Perhaps the mechanisms I describe above fit already into the schemes outlined by the authors, but none of this is made explicit. There needs to be stronger connection to the physiology and body temperatures of moths in their environments. (B) I was surprised to see no discussion of diurnal patterns of activity and flight, given that most (but not all) moths are night-flying.

Authors' responses 1:

We thank the reviewer sincerely for their thoughtful and supportive comments, especially regarding the originality and ambition of our modeling and empirical approach. We deeply appreciate the insightful points raised about the nuanced relationship between ambient environmental temperatures and the actual thermal conditions experienced by moths, particularly given their capacity for shivering endothermy during nocturnal flight.

We agree that moths' ability to elevate thoracic temperatures during activity introduces important complexity when interpreting temperature exposure and

its selective effects. In response, we have added new discussion to more explicitly consider the physiological realities of moth thermal biology. Specifically, we clarify that although many moths thermoregulate actively during flight, selective pressures are more likely to act during periods of inactivity—when moths and other life stages remain subject to ambient thermal variability. We also highlight the role of immobile life stages and microhabitat conditions in shaping thermal selection, especially in topographically complex montane environments.

new discussion paragraph:

“Most sampled moths are nocturnal fliers that rely on shivering endothermy to elevate thoracic temperatures to approximately 33–38 °C³³, thereby buffering operative temperature fluctuations during activity. However, selective pressures on thermal physiology are likely to act primarily during periods of inactivity. Although adult moths typically seek shaded refugia during daylight hours, immobile life stages remain exposed to fine-scale microclimatic gradients shaped by topography and solar input. Microclimatic thermal heterogeneity exhibits significant three-way interactions among region, diel period, and elevation (Supplementary Material). Across multiple regions and time periods, high-elevation environments consistently show greater thermal heterogeneity³⁶, with daytime variability exceeding that of nighttime. Previous studies further indicate that in mountainous systems, heterogeneity is strongly shaped by topographic roughness, canopy architecture, and spatiotemporal variation in precipitation³⁶. In the montane cloud forests of Taiwan, fog and low cloud layers exert powerful radiative control over the diurnal temperature range (DTR), producing nonlinear elevational gradients and marked discontinuities at mid- to high elevations³⁷. These discontinuities align with zones of persistent fog and highlight the complex spatial structure of thermal regimes in montane ecosystems. Within this framework, environmental favorability—defined by the confluence of benign temperatures and high primary productivity—emerges as a principal driver of thermal trait diversity. Warm, productive environments support both thermal specialists and generalists, expanding trait hypervolume and increasing assemblage-level variability. By contrast, although high-elevation habitats exhibit substantial microclimatic variation, their overall thermal severity imposes strong environmental filtering that compresses trait distributions. Greater DTR may elevate CTmax in sedentary stages and increase average TTrange, yet this occurs without corresponding expansion in trait hypervolume. These findings support the idea that even short-term fluctuations, occurring on timescales shorter than species’ generation lengths, can shape evolutionary outcomes distinct from those driven by

long-term means. However, local environmental conditions—averaged over biologically relevant timescales—may exert stronger selective pressures than those reflected in long-term climatic averages alone³⁵.” (L303-325)

Reviewer's comments 2:

2. My second point is related to the first in the sense that it's another process that can decouple drivers of moth body temperatures from 'ambient temperatures.' I'm referring here to microclimates, which have received a lot of attention recently in the literature on insects (and other small ectotherms). The idea is that coarse measures of air temperature in space and time can do a poor job of capturing thermal experiences of actual insects, which tend to be much more diverse. I.e., there is typically a lot of thermal diversity across space and time that is not accounted for by data loggers and mean air temperatures. It's possible that this problem would be most acute for animals operating in the daytime, during which differences in sun and shade can drive large differences in operative temperatures at very fine spatial scales. This may be less of a concern for night-active animals like moths, as nighttime microclimatic temperatures probably are less diverse. Nevertheless, moths live also through daytimes, and likely experience quite substantial variation in daytime temperatures experienced – and this variation may differ according to elevation or species. For example, if lowlands are sunnier on average than highlands, microclimatic diversity in operative temperatures may be on average higher in lowlands. If individual species use this diversity on average in different ways, then long-term evolution of assemblages could lead to patterns like those seen in Figure 4d. Here too, maybe this is just a more mechanistic description of what the authors are suggesting already, but it's not clear to me that this is the case – plus I was surprised to see nothing about microclimates mentioned anywhere in the manuscript.

Authors' responses 2:

We thank the reviewer for this important point regarding the role of microclimates in shaping insect thermal experiences. We agree that coarse-scale temperature metrics can poorly reflect the actual thermal environments experienced by small ectotherms, especially given the fine-scale spatial and temporal variability inherent in natural habitats.

In response, we have explicitly incorporated discussion of microclimatic heterogeneity in both the main text (Discussion) and Supplementary Results.

We now address how microclimatic gradients—especially those driven by topography, canopy structure, and solar input—may create diverse thermal exposures, particularly during the daytime. While nocturnal moths may experience less thermal variability during their active periods, their immobile daytime phases and other life stages (e.g., pupae, larvae) are exposed to and potentially shaped by this thermal heterogeneity.

Importantly, we analyzed and reported diel patterns of microclimatic heterogeneity across regions and elevations, revealing consistent patterns of greater variability during the day than at night. These additions help clarify that the observed assemblage-level patterns could emerge from microclimatic filtering, particularly under conditions where daytime thermal variation is substantial even at higher elevations.

Relevant new discussion:

“Microclimatic thermal heterogeneity exhibits significant three-way interactions among region, diel period, and elevation (Supplementary Material). Across multiple regions and time periods, high-elevation environments consistently show greater thermal heterogeneity³⁶, with daytime variability exceeding that of nighttime. Previous studies further indicate that in mountainous systems, heterogeneity is strongly shaped by topographic roughness, canopy architecture, and spatiotemporal variation in precipitation³⁶. In the montane cloud forests of Taiwan, fog and low cloud layers exert powerful radiative control over the diurnal temperature range (DTR), producing nonlinear elevational gradients and marked discontinuities at mid- to high elevations³⁷. These discontinuities align with zones of persistent fog and highlight the complex spatial structure of thermal regimes in montane ecosystems. Within this framework, environmental favorability—defined by the confluence of benign temperatures and high primary productivity—emerges as a principal driver of thermal trait diversity. Warm, productive environments support both thermal specialists and generalists, expanding trait hypervolume and increasing assemblage-level variability. By contrast, although high-elevation habitats exhibit substantial microclimatic variation, their overall thermal severity imposes strong environmental filtering that compresses trait distributions. Greater DTR may elevate CTmax in sedentary stages and increase average TTrange, yet this occurs without corresponding expansion in trait hypervolume.” (L307-322)

Relevant new results:

“Microclimate heterogeneity (i.e., temperature variability) showed significant three-way interaction effects among region, period, and elevation (LMM, Region × Period ×

Elevation, $P < 0.001$; Supplementary Fig. 16; Supplementary Table 7). Across all elevations and periods, Mt. Jiabin exhibited the highest temperature heterogeneity, followed by Cameron Highlands and then Mt. Hehuan ($P < 0.001$ for all post hoc comparisons; Supplementary Fig. 16; Supplementary Table 8). Daytime heterogeneity consistently exceeded nighttime heterogeneity in all regions ($P < 0.001$; Supplementary Fig. 16; Supplementary Table 9).

Elevational trends differed among regions and diel periods. In Mt. Jiabin, temperature heterogeneity increased significantly with elevation during both day and night ($P < 0.001$ for both; Supplementary Fig. 16a; Supplementary Table 10). In Mt. Hehuan, daytime heterogeneity also increased with elevation ($P < 0.001$), whereas nighttime heterogeneity showed no significant trend ($P = 0.08$; Supplementary Fig. 16b; Supplementary Table 10). In Cameron Highlands, neither daytime nor nighttime heterogeneity exhibited significant elevational patterns ($P > 0.1$ for both; Supplementary Fig. 16c; Supplementary Table 10).” (Supplementary Discussion, L341-352)

We hope these additions address the reviewer’s insightful observation and help clarify the ecological and mechanistic relevance of microclimates to our findings.

Reviewer’s comments 3:

3. My other complaint is about how the data and analyses are presented. Overall, the approach is to present quite abstract and derived version of the data (both modeled and empirical) – things like hypervolumes and K-mean cluster distributions. In addition, the basic data on temperature tolerances are rendered as TTRange (thermal tolerance range) and the standard deviation of TTRange. These latter two are more approachable, but they’re still derived. For example, I wondered whether CTmax or CTmin changes more with the various temperature measurements. I think I can figure that out but nowhere are there plots of each of those things individually against the independent variables.

Authors’ responses 3:

Thank you for this valuable feedback regarding data presentation. We agree that examining the underlying components of thermal tolerance is essential for understanding the mechanisms that shape assemblage-level patterns. In

response, we have added new analyses that evaluate CTmax and CTmin individually.

As now described in the manuscript:

“We further examined how environmental variables affect CTmax and CTmin (Supplementary Fig. 13). Using SHapley Additive exPlanations (SHAP) analysis (see Methods for details), we identified that CTmax is primarily driven by diurnal maximum temperature (DTmax), while CTmin is most strongly influenced by diurnal mean temperature (DTmean) (Supplementary Fig. 13a, b). These relationships were confirmed through linear regressions, showing significant positive associations between CTmax and DTmax ($\beta = 0.42$, $P < 0.001$; Supplementary Fig. 13g), and between CTmin and DTmean ($\beta = 0.44$, $P < 0.001$; Supplementary Fig. 13i).” (L246-252)

These findings clarify that CTmax and CTmin respond to distinct environmental drivers, offering mechanistic insight into how individual trait responses contribute to the broader assemblage-level patterns.

While our primary analyses still focus on emergent properties such as trait hypervolume and clustering—since these capture community-level patterns not visible at the species level—we believe that adding this disaggregated view of thermal limits provides valuable context. The combination of both approaches allows readers to examine thermal adaptation across multiple levels of biological organization.

Reviewer's comments 4:

Likewise for temperature data – like it would be nice to have max and min temperatures for the different sites plotted separately, perhaps against altitude.

I understand that the current approach is more compact in that it summarizes lots of information within single parameters, but the cost is that it's more difficult to understand what's happening within the raw data to drive those changes.

Authors' responses 4:

Thank you for this suggestion to include more direct visualization of the underlying temperature data. We agree that providing raw environmental context helps readers interpret the patterns observed in thermal trait diversity.

In response, we have added new analyses that show how basic temperature variables vary with elevation across our sites. As described in the revised manuscript:

“To improve our understanding of the thermal conditions experienced by species across the elevation gradient, we plotted the relationships between annual mean temperature, STR and DTR against elevation (Supplementary Figure 15). As expected, mean temperature decreased significantly with elevation, while STR increased and DTR remained relatively stable.” (L258-261)

These additions directly address the reviewer’s request by illustrating how maximum and minimum temperatures, as well as their derived variability metrics, change across the gradient. While our primary analyses continue to focus on assemblage-level summaries such as trait hypervolumes, these added plots offer a clearer view of the raw environmental variation that drives trait patterns.

We believe this dual-level presentation—linking basic environmental variables to emergent ecological outcomes—enhances interpretability without sacrificing the generality of our framework.

Reviewer's comments 5:

Line 233: technically the measurements of wild moths do not constitute ‘thermal performance curves’ but rather tolerance ranges or critical thermal limits.

Authors' responses 5:

Modified to “critical thermal limits” as suggested.

Reviewer's comments 6:

Reviewer #2 (Remarks to the Author):

The article combines modelling and field data from moth species in Asia, across an elevation gradient, to test predictions derived from three selected hypotheses. The hypotheses aim at explaining biodiversity patterns in thermal adaptations, and were named: favorability (F), long-term seasonal variation (LT), and short-term daily fluctuations (ST). The results support F in the context of a tropical region, which, citing the article words (line 235), "aligns with classical theory". Thus, I would recommend to clearly frame the novelty of the study, beyond referencing the scale of the study (line 257). However, I have concerns regarding (i) conceptual foundations and the writing of the

manuscript, and (ii) model communication and implementation. Below I summarize my concerns and provide suggestions for improvements. However, the list is not exhaustive as my decision is to not recommend this article for publication in Nature Communications journal.

Authors' responses 6:

Thank you for this thoughtful comment. We have clarified the novelty of our approach, which goes beyond data scale to introduce a mechanistic framework that integrates eco-evolutionary modeling with empirical trait data across multiple temporal scales. To our knowledge, this is the first study to directly compare the roles of short-term (daily) and long-term (seasonal) thermal variability in shaping functional trait diversity at the assemblage level.

Our study addresses a fundamental distinction between species-level and assemblage-level patterns. As we clarify in the introduction:

"At the species level, thermal specialization refers to the breadth of a single species' thermal-tolerance range ($TTrange = CTmax - CTmin$); narrow ranges indicate specialists, while wider ranges indicate generalists. In contrast, assemblage-level diversity in thermal-adaptation strategies is quantified as the n-dimensional hypervolume formed by all species' $CTmin-CTmax$ combinations. As such, an assemblage may simultaneously exhibit high specialization (many narrow-range species) and high hypervolume (a wide collective niche space) when specialists occupy distinct regions of trait space." (L69-75)

Mechanistically, our model demonstrates that benign environments support greater diversity in thermal strategies through two simultaneous processes:

"In our framework, the mechanisms shaping thermal trait composition arise from two simultaneous processes: environmental filtering imposed by thermal regimes, and biotic interactions such as interspecific competition. The former determines which thermal strategies are viable, while the latter regulates the abundance and success of those strategies within assemblages." (L104-107)

This reveals that warmer lowlands reduce abiotic filtering, allowing both specialists and generalists to coexist, rather than simply pushing species toward generalism alone. We have also substantially revised the

supplementary materials to better explain our model assumptions and strengthen connections between thermal trait patterns and moth biology. This expands classical favorability theory into the realm of functional diversity and physiological trait space, providing a predictive framework for understanding how climate change may reshape thermal adaptations across different ecological systems.

Reviewer's comments 7:

(i) conceptual foundations and the writing of the manuscript

The writing feels rush. However, papers are messages and I would recommend to take the time to frame, write and deliver the messages such that they contribute to progress in the way intended by the authors. To this work, the modelling part is as important as the field work. However, model description could benefit from appropriate justification of modelling decisions, and from explaining the terminology and acronyms being used (section (ii) expands on this issue).

Authors' responses 7:

Thank you for raising this important concern. In response, we have substantially revised both the main text and supplementary materials to improve clarity, narrative flow, and conceptual framing—especially in the model sections.

We now provide more explicit justifications for key modeling choices, such as our use of immigration over mutation in the baseline model:

"In our main model, we simulate the introduction of new species via immigration, analogous to allopatric speciation. This approach was chosen because it facilitates efficient exploration of the thermal trait space and accelerates the system's convergence to evolutionary steady states, particularly under fluctuating environmental conditions." (Supplementary Methods, L189-192)

We have also clarified the terminology used in the manuscript. The revised supplementary materials now open with precise definitions of core model components:

"To maintain consistency with terminology used in the main text, we define each biological unit in the model across spatial and organizational scales. A patch is a spatial unit where individual organisms reside; an individual is the basic entity

subject to biological processes such as dispersal, reproduction, and mortality. A species is composed of individuals that share the same thermal performance curve, defined by three trait parameters. An assemblage refers to the full set of species with at least one surviving individual across all patches at a given time point."
(Supplementary Methods, L27-32)

Reviewer's comments 8:

The article says that the cited hypotheses are competing hypotheses. However, this is not clear to me. Actually, observations on patterns of thermal adaptations of species could arise due to more than to single reasons/hypotheses. Eg., F and LT could get closer to what is expected from "constant" environments, which would lead to reduced plasticity and higher specialization (I recommend to check the literature on phenotypic plasticity theory, many are based on individual-based models). Further, if the life cycle of the species in question is longer than the environmental fluctuations (eg., under ST, which is the case for the moths), the mean environment could strongly lead the eco-evolutionary dynamics. I've seen that the study has come across this in lines 228-230.

Authors' responses 8:

Thank you for this insightful and conceptually rich comment. We have revised the introduction and methods to clarify our framing of the hypotheses and their potential interdependence.

In particular, we no longer refer to these as "competing hypotheses," and instead describe them as alternative mechanisms that generate distinct and testable predictions but may operate in tandem depending on ecological and temporal context. As revised in the manuscript:

"Three alternative hypotheses have been proposed to explain thermal trait patterns across environmental gradients, each offering distinct and testable predictions. These mechanisms—favorability, long-term seasonal variation, and short-term daily fluctuation—are not mutually exclusive and may jointly influence trait evolution depending on the ecological context and temporal scale." (L30-33)

We agree with your point that trait patterns often emerge from multiple interacting drivers. For instance, the effects of favorability and long-term predictability may both contribute to increased specialization in stable

environments, consistent with classical phenotypic plasticity theory. However, our individual-based model suggests that in productive, low-filtering environments, diversity increases not through convergence on specialization, but via the coexistence of both narrow- and broad-range species. As clarified in the supplementary discussion:

“Although our model does not incorporate within-generation phenotypic plasticity, the distinction between plastic responses and long-term trait evolution remains conceptually important. Classical plasticity models^{61,62} focus on how environmental predictability favors plasticity, whereas our framework explores the selection of fixed thermal traits. Future integration of these perspectives may offer broader insights into physiological adaptation under environmental variability.” (Supplementary Discussion, L284-289)

Your observation about the relationship between life cycle length and environmental fluctuation timescales is also crucial. While moth generation times do exceed the duration of daily fluctuations, our results demonstrate that short-term variability can still select for distinct strategies. We have clarified this point:

“The distinction between short-term and long-term variability effects represents an important advance in our understanding of thermal adaptation mechanisms, because it highlights that environmental fluctuations at different temporal scales can select for fundamentally different thermal strategies. While long-term variability tends to favor generalists with broad tolerance ranges, short-term fluctuations may support specialists adapted to transient favorable conditions. Recognizing this separation allows for more precise predictions about how organisms respond to climate dynamics, especially in the face of increasing environmental variability under global change.” (L57-63)

This modeling assumption is grounded in the natural history of our focal system, as now described in the revised introduction:

“The developmental rate of moths changes markedly with climate. Under warm conditions, many species complete their entire life-cycle from egg to adult in about 6–10 weeks, whereas at higher latitudes or in cooler environments a generation can stretch to 3–4 months, with some species even requiring overwintering before eclosion. By contrast, the adult stage is typically very brief—most noctuid moths live only 7–14 days³⁰. Accordingly, our model assumes that organismal lifespan exceeds short-term (e.g., daily) fluctuations but is shorter than long-term seasonal variation,

enabling us to test how environmental variability across timescales influences the evolution of thermal traits.” (L85-91)

We appreciate your encouragement to engage more deeply with these foundational theories and believe the revised manuscript reflects a clearer and more integrated conceptual foundation.

Reviewer's comments 9:

In the abstract and the main text, there is mentioning and discussion on the mechanisms shaping thermal adaptation. However, there is little if any, mention to evolutionary mechanisms. Thus, what is meant by mechanisms in the context of this study? I expect that understanding the mechanisms means understanding the why the patterns we observe, which would be very interesting. However, I do not see that here.

Authors' responses 9:

Thank you for this important question. We have clarified our use of the term “mechanisms” throughout the manuscript to better reflect the ecological and evolutionary processes that shape thermal trait distributions within assemblages.

In our framework, mechanisms refer to the processes that determine which thermal-performance curves persist under given environmental conditions. These include both:

- Environmental filtering, which sets the viability of trait strategies based on local thermal regimes
- Biotic interactions, particularly interspecific competition, which shape the relative abundance of viable strategies

We now define this explicitly in the model description:

“In our framework, the mechanisms shaping thermal trait composition arise from two simultaneous processes: environmental filtering imposed by thermal regimes, and biotic interactions such as interspecific competition. The former determines which thermal strategies are viable, while the latter regulates the abundance and success of those strategies within assemblages.” (L104-107)

Environmental filtering operates by influencing species fitness through temperature-dependent survival and reproduction, effectively setting the outer boundaries of the trait space. Within these boundaries, competition further structures trait distributions by filtering out less fit species in resource-limited systems.

To more directly address the role of evolutionary mechanisms, we developed an additional variant of the model in which new thermal strategies emerge via mutation from existing species, simulating a simplified form of local diversification akin to sympatric speciation. This complements our baseline model, which assumes diversity arises through immigration from a regional pool.

As described in the revised text:

“To test the robustness of this assumption, we implemented an alternative mechanism for generating diversity based on sympatric speciation via mutation. In this variant, new species arise from existing species at the same rate as immigration... This alternative produced consistent results across the three environmental hypotheses, indicating that our conclusions are not sensitive to the mode of diversity generation.” (Supplementary Methods, L201-239)

These findings are presented in Supplementary Figs. 10–12. We also now highlight how interspecific competition shapes the final trait composition:

“Finally, we reran the simulations with and without interspecific competition. These comparisons revealed a two-step process: (i) environmental filtering imposed by the thermal regime, determining whether a species can persist, followed by (ii) interspecific competition, which further narrows the realized trait distribution through differential reproductive success.” (L208-212)

Together, these revisions clarify what we mean by "mechanisms" in this study and emphasize their grounding in both ecological dynamics and evolutionary processes.

Reviewer's comments 10:

lines 5-7 (and elsewhere): what is the difference between thermal specialization and great diversity in thermal adaptation strategies? For

instance, an assemblage of species with a high diversity of thermal adaptation strategies can result from being composed out of many specialist...

Authors' responses 10:

Thank you for this important clarification request. We agree that the distinction between species-level thermal specialization and assemblage-level diversity in thermal adaptation strategies warrants explicit explanation, as they represent fundamentally different ecological scales.

At the species level, thermal specialization refers to the breadth of a single species' thermal tolerance range ($T_{\text{range}} = CT_{\text{max}} - CT_{\text{min}}$); species with narrower ranges are considered thermal specialists, while those with broader ranges are generalists. In contrast, assemblage-level diversity is quantified as the n -dimensional hypervolume formed by the full set of $CT_{\text{min}}-CT_{\text{max}}$ combinations across all species in a community.

We have clarified this distinction in the revised introduction:

“At the species level, thermal specialization refers to the breadth of a single species' thermal-tolerance range ($T_{\text{range}} = CT_{\text{max}} - CT_{\text{min}}$); narrow ranges indicate specialists, while wider ranges indicate generalists. In contrast, assemblage-level diversity in thermal-adaptation strategies is quantified as the n -dimensional hypervolume formed by all species' $CT_{\text{min}}-CT_{\text{max}}$ combinations.” (L69-73)

As the reviewer rightly notes, these two forms of diversity are not mutually exclusive: an assemblage can simultaneously exhibit high specialization (many species with narrow ranges) and high trait hypervolume, provided those specialists occupy distinct regions of trait space. We have added the following clarification to the introduction:

“As such, an assemblage may simultaneously exhibit high specialization (many narrow-range species) and high hypervolume (a wide collective niche space) when specialists occupy distinct regions of trait space.” (L73-75)

This distinction is central to our findings. In warmer environments, we observe both greater representation of thermal specialists and an expansion of assemblage-level trait space. This suggests that favorability supports coexistence of diverse strategies, rather than promoting generalism alone.

Reviewer's comments 11:

line 26: emergent properties are a characteristic of the agent-based (also called individual-based) approach. Therefore, species level studies can potentially display emergent properties as well, depending on how they are modeled.

Authors' responses 11:

Thank you for this correction. We agree that emergent properties can arise from any individual-based (agent-based) model, including those focused on species-level dynamics, depending on the structure and interactions specified in the model.

Our intended meaning was more specific: the particular emergent property we focus on—the expansion or contraction of functional trait space at the assemblage level—only becomes apparent when traits are aggregated across co-occurring species. While individual species analyses yield tolerance curves or trait values, the broader shape and structure of the community's trait hypervolume is a property of the assemblage as a whole.

We have clarified this distinction in the main text:

“The strength of assemblage-level analyses is that they capture emergent properties—attributes of the whole assemblage that cannot be inferred by inspecting species in isolation. For example, coral assemblages recover only 60% of their original trait space after extreme heat waves¹⁴, a pattern undetectable through individual species analyses¹⁵⁻¹⁷.” (L21-24)

We appreciate the reviewer's point and have revised the language to more clearly describe the level at which emergent properties arise in our specific analysis.

Reviewer's comments 12:

line 57: why such distinction represents an advance to our understanding?
Please clarify

Authors' responses 12:

We have clarified this advance in the introduction:

“The distinction between short-term and long-term variability effects represents an important advance in our understanding of thermal adaptation mechanisms, because it highlights that environmental fluctuations at different temporal scales can select for fundamentally different thermal strategies. While long-term variability tends to favor generalists with broad tolerance ranges, short-term fluctuations may support specialists adapted to transient favorable conditions^{16,22,23}. Recognizing this separation allows for more precise predictions about how organisms respond to climate dynamics, especially in the face of increasing environmental variability under global change.”
(L57-63)

Reviewer's comments 13:

line 65: "has never" Drop this, and rather focus on the uniqueness and potential of the study

Authors' responses 13:

Modified as suggested. The revised text in the introduction now reads:

“However, how these three hypotheses and their underlying mechanisms affect thermal functional trait diversity at the assemblage level has not been directly studied within a comprehensive framework.” (L75-77)

Reviewer's comments 14:

lines 71-75: include here an statement of what is known about the diversity of moths. Are they more diverse in lower variable - tropical - environments than in "highly" fluctuating temperate ones? Also, information on the life cycle is needed. To my knowledge moth's life cycle is days or months. Not a single day. How does this fact affect the modelling of the daily vs long-term fluctuations?

Authors' responses 14:

Thank you for this important point about moth diversity patterns and life cycle timing. We have revised this section to include relevant natural history information and explain how it supports our modeling assumptions.

As now stated in the introduction:

“Moths often exhibit remarkable species diversity in thermally stable tropical forests—for example, a single 16-km² Andean cloud-forest plot supports over 1,100 geometrid species, more than 6% of global geometrid diversity²⁹. The developmental rate of moths changes markedly with climate. Under warm conditions, many species complete their entire life-cycle from egg to adult in about 6–10 weeks, whereas at higher latitudes or in cooler environments a generation can stretch to 3–4 months, with some species even requiring overwintering before eclosion. By contrast, the adult stage is typically very brief—most noctuid moths live only 7–14 days³⁰. Accordingly, our model assumes that organismal lifespan exceeds short-term (e.g., daily) fluctuations but is shorter than long-term seasonal variation, enabling us to test how environmental variability across timescales influences the evolution of thermal traits.” (L83-91)

Based on this, our model assumes that organismal lifespan is **longer than short-term (e.g., daily) fluctuations** but **shorter than long-term seasonal variation**, which reflects the real-world timing of moth development and thermal exposure. This allows us to investigate how environmental variability across timescales influences thermal trait evolution in organisms with intermediate life histories like moths.

Reviewer's comments 15:

lines 122-124: to what extent could this result from model assumptions? ie., soft vs abrupt decays in thermal tolerance below vs above optimal temperature (S4), plus the imposed upper temperature maximum limit (S1)

Authors' responses 15:

Thank you for this thoughtful comment. We agree that assumptions about the shape of thermal performance curves—including asymmetry around the optimum and the position of an upper temperature limit—could influence model outcomes. To assess this, we conducted additional analyses to test the robustness of our results.

First, to evaluate the influence of curve shape, we implemented a model variant in which the area under each species' thermal performance curve—previously held constant—was instead rescaled using random values from a normal distribution. This introduced naturalistic variation in performance

height and width, allowing both broader generalist and narrower specialist curves. As described in Supplementary Fig. 1:

“The main model assumes all species have identical areas of thermal performance curves (i.e., a strong correlation between the width and height). We relaxed this assumption here by rescaling the area with a random number from a normal distribution, with mean of 1 and standard deviation of 0.1 (a) or 0.25 (b).”
(Supplementary Figures, L358-361)

Crucially, this modification did not alter the main result: we continued to observe increased trait hypervolume in warmer environments, indicating that the pattern is not an artifact of constrained curve geometry.

Second, we tested whether the observed hypervolume expansion could result from curve asymmetry alone by removing the productivity-related mechanism from the model. As shown in Supplementary Fig. 2:

“Model results with no productivity assumption. Hypervolume of thermal traits against mean ambient temperature ($\beta = -0.016$, $P = 0.591$, $R^2 = 0.03$).”
(Supplementary Figures, L365-369)

This confirmed that trait diversification does not arise from thermal performance curve asymmetries, but rather from eco-evolutionary dynamics under relaxed environmental filtering in more favorable (warmer) conditions.

Taken together, these tests show that while the shape of the thermal performance curve can affect individual species limits (e.g., CTmin and CTmax), our key result—assemblage-level expansion of thermal trait diversity in warmer environments—is robust to multiple curve configurations and is not driven by modeling artifacts.

Reviewer’s comments 16:

lines 127-132: Can one discard that there are not two extreme strategies in the assemblage that result in high variance. Thus, no diversity in strategies. Just high variation between few strategies.

Authors’ responses 16:

Thank you for raising this important point. We agree that high trait variance, by itself, does not necessarily imply diversity in thermal strategies. It could, in

theory, result from a small number of extreme values. However, our analyses indicate that this is not the case.

First, both our model and empirical data show that high variance in TTrange is accompanied by increased thermal trait hypervolume (Fig. 4b). If the variance were driven by just two extreme strategies, the hypervolume would remain limited due to large portions of trait space being unoccupied. Instead, we observe an expansion across trait space, consistent with the coexistence of multiple distinct strategies.

Second, analyses of species composition (Fig. 5b) demonstrate gradual shifts in the proportion of warm-adapted, cold-adapted, heat-tolerant, and heat-sensitive species across temperature gradients, rather than a bimodal structure. This indicates continuous variation rather than dominance by a few extremes.

Finally, clustering results (Supplementary Table 4) further support the presence of multiple thermal strategies in the assemblages, rather than high variation driven by a few outliers.

Together, these patterns confirm that the observed trait variance reflects genuine ecological diversity in thermal adaptation strategies.

Reviewer's comments 17:

(ii) Model communication and implementation

The main text will benefit very much from having precise information and justification regarding model selection, assumptions, decisions, and explanations - e.g., regarding model terminology, crucial: what is an assemblage to the model? The supporting information needs considerable work with the order, sequence and drafting in general. I would suggest to find a good modelling paper to use as a reference. Particularly, when deciding for parameter names and subscripts. At the moment is all very confusing. As I mentioned above, the modelling is an important step in this study, and deserves to be conveyed as such.

Authors' responses 17:

Thank you for this important comment. We agree that clear communication of the model framework is essential, particularly given the central role modeling

plays in our study. We have made substantial revisions to improve the clarity and accessibility of both the main text and the supplementary materials.

First, we clarified key terms and structure in the supplementary methods, including a precise definition of “assemblage”:

“To maintain consistency with terminology used in the main text, we define each biological unit in the model across spatial and organizational scales. A patch is a spatial unit where individual organisms reside; an individual is the basic entity subject to biological processes such as dispersal, reproduction, and mortality. A species is composed of individuals that share the same thermal performance curve, defined by three trait parameters. An assemblage refers to the full set of species with at least one surviving individual across all patches at a given time point.”

(Supplementary Methods, L27-32)

We use “assemblage” rather than “community” to emphasize that our model tracks groups of co-occurring species exposed to the same environmental conditions, without assuming strong or persistent ecological interactions among all species. This terminology aligns with common practice in trait-based and macroecological studies, where functional trait space is measured across regional pools or assemblages without requiring constant direct interaction among species.

Second, we revised parameter naming conventions throughout the supplementary materials to improve consistency and readability. For example:

Original	Revised	Notes
T	$T_{current}$	Clarifies this is the current temperature
$fecundity$	f_{env}	Uses ‘f’ to represent fitness-related parameter
pf	$f_{spc,T}$	Indicates the focal species’ current thermal performance value
$weight$	w	Simplification to improve the format consistency
$Popn_{i,j}$	$n_{i,j}$	Simplification to improve the format consistency
$comp_{i,j}$	$f_{spc,i}$	Clarifies this is the focal species’ density-dependent fitness at patch i

Lastly, we reorganized the supplementary methods to follow a more logical structure—moving from environmental inputs to species dynamics and then to trait and assemblage-level outputs. These changes are intended to make the modeling framework easier to navigate and more transparent for readers unfamiliar with individual-based modeling.

We appreciate the recommendation to consult reference modeling papers, and have adopted their conventions where applicable. We hope these revisions make the modeling elements of the manuscript clearer and more accessible.

Reviewer's comments 18:

Supplementary text

The eco-evolutionary model

- The section "overview" needs to start with a clear motivation and objectives of the modelling.

Also, model selection and decisions need to be justified and model concepts, traits and measurements, clearly explained. E.g., it is not known what an assemblage is (to the model) until the very end (and still its definition is not precise). This needs to be at the beginning. Is the model spatially explicit or implicit? what are the basic entities considered "individuals" in the model? the species?

how do "individuals" differ? in what traits?

Authors' responses 18:

We have restructured the overview section of the supplementary methods to begin with clear motivation and model objectives, followed by precise definitions of key concepts. The revised overview now starts with:

"Our model simulates eco-evolutionary dynamics by tracking how thermal fluctuations and species interactions determine which thermal strategies persist within a structured population. Species were generated with randomly sampled thermal performance traits and distributed across multiple patches, each initialized with potentially different species compositions." (Supplementary Methods, L18-21)

We have added explicit definitions at the beginning of the methods section:

"To maintain consistency with terminology used in the main text, we define each biological unit in the model across spatial and organizational scales. A patch is a spatial unit where individual organisms reside; an individual is the basic entity subject to biological processes such as dispersal, reproduction, and mortality. A species is composed of individuals that share the same thermal performance curve, defined by three trait parameters. An assemblage refers to the full set of species with at least one surviving individual across all patches at a given time point."
(Supplementary Methods, L27-32)

We have also clarified the spatial structure and model framework:

"While all patches experience the same temperature at each time point, this structured population setup mimics the mosaic of local habitats in nature and allows species to persist locally even when conditions become temporarily unfavorable elsewhere. This implicit spatial structure helps buffer against the effects of short-term demographic stochasticity, enabling more realistic patterns of species survival over time." (Supplementary Methods, L21-25)

The model objectives and conceptual framework are now clearly presented before diving into the technical implementation details.

Reviewer's comments 19:

- Clearly define what is considered an assemblage in the context of the model. It could be fair (to the reader) to assume that each patch is an assemblage of species; while later an assemblage is barely defined as some consideration of all species across all patches... This definition needs to be crystal clear.

Authors' responses 19:

Indeed, we have addressed this in the revised overview section (see previous response).

Reviewer's comments 20:

- Furthermore, it feels that evolution is lacking even given that it is said to be an eco-evolutionary model. To me it seems that the model is purely ecological

with emigration/immigration dynamics. As it is at the moment, I would suggest to drop "eco-evolutionary" from the name.

Authors' responses 20:

Thank you for this thoughtful comment. We agree that a clear justification is needed when referring to a model as "eco-evolutionary." Our original implementation used an immigration-based framework, in which species with novel thermal traits entered the system from an external pool. This approach mimics allopatric speciation, allowing trait turnover through the continual introduction of species with new thermal strategies.

To directly address the concern about endogenous evolution, we conducted an additional modeling experiment incorporating sympatric speciation via mutation. In this version, new species arise from existing ones within the assemblage by inheriting parental traits with added Gaussian variation. As described in the supplementary methods:

"To test the robustness of this assumption, we implemented an alternative mechanism for generating diversity based on sympatric speciation via mutation. In this variant, new species arise from existing species at the same rate as immigration... The key difference lies in how thermal traits are assigned. Instead of independent sampling as in the immigration design, the new species inherits its trait structure from a randomly chosen parent species, with variation introduced by Gaussian noise." (Supplementary Methods, L201-206)

Importantly, this mutation-based model produced results consistent with those of the immigration-based model:

"This alternative produced consistent results across the three environmental hypotheses, indicating that our conclusions are not sensitive to the mode of diversity generation." (Supplementary Methods, L237-239)

This outcome supports the robustness of our findings and demonstrates that the observed trait diversification arises from the interaction between ecological filtering and trait evolution, regardless of whether diversity originates from immigration or mutation.

For this reason, we retain the use of “eco-evolutionary” to describe our framework, which integrates ecological processes (e.g., competition, filtering) with evolutionary dynamics (e.g., trait diversification via mutation or speciation).

Reviewer's comments 21:

- Then, why to assume explicit space, ie., many patches, instead of implicit space with N_t species defining the assemblage? this is not explained. Wouldn't that be simpler? Please justify

Authors' responses 21:

Thank you for this insightful question. We chose to use a structured population model with multiple patches, rather than a single well-mixed assemblage of N_t species, to better approximate the dynamics of natural systems.

This decision was motivated by two key considerations:

1. **Ecological realism:** In nature, species often experience competition, survival, and reproduction at local spatial scales. A patch-based structure allows the model to reflect this ecological heterogeneity—capturing localized species dynamics and the mosaic-like distribution of habitats typical of many landscapes.
2. **Demographic buffering:** Structured populations allow species to persist in some patches even when conditions in other patches become temporarily unsuitable. This spatial configuration provides resilience against stochastic extinctions caused by short-term environmental fluctuations, mimicking natural “rescue effects” that are especially relevant under variable climates.

We have clarified this rationale in the supplementary methods:

“While all patches experience the same temperature at each time point, this structured population setup mimics the mosaic of local habitats in nature and allows species to persist locally even when conditions become temporarily unfavorable elsewhere. This implicit spatial structure helps buffer against the effects of short-term demographic stochasticity, enabling more realistic patterns of species survival over time.” (Supplementary Methods, L21-25)

In sum, while a simpler, non-spatial model could be implemented, it would limit our ability to capture the biologically relevant processes of localized extinction and persistence, which are especially important for testing trait evolution under fluctuating environmental conditions.

Reviewer's comments 22:

- Why to use migrations (potentially increasing species number) and not mutations, which leads to variations of existing species? Please justify. Further, how migration rate is adjusted to fit the different time scales? for example, considering daily fluctuations, moths immigrants per day would be expected to be much lower than moth immigrants per days or months.

Authors' responses 22:

Thank you for this comment. We chose to use immigration as our initial mechanism for introducing new thermal strategies because it efficiently explores trait space and simulates allopatric speciation—where new species arise from outside the focal system. This approach enables rapid sampling of diverse thermal traits under different environmental regimes.

To directly address concerns about evolutionary realism, we also implemented an alternative model incorporating sympatric speciation via mutation. In this version, new species arise from existing ones through random variation in inherited traits. As described in the supplementary methods:

“To test the robustness of this assumption, we implemented an alternative mechanism for generating diversity based on sympatric speciation via mutation... The new species inherits its trait structure from a randomly chosen parent species, with variation introduced by Gaussian noise.” (Supplementary Methods, L201-206)

Crucially, both the immigration- and mutation-based models produced consistent results across all environmental hypotheses:

“This alternative produced consistent results across the three environmental hypotheses, indicating that our conclusions are not sensitive to the mode of diversity generation.” (Supplementary Methods, L237-239)

Regarding your second question on temporal scaling: we agree that, in real systems, daily immigration of moth species would be rare. However, our model is not meant to directly map each time step onto real-world days. As clarified in the supplementary methods:

“Our model is a heuristic framework that captures environmental fluctuations across multiple time scales, but it is not intended for direct translation of a single time step into a real-world unit such as a day.” (Supplementary Methods, L32-34)

Each time step aggregates demographic processes such as birth, death, and species turnover. This abstraction allows us to use a fixed immigration (or mutation) rate across all scenarios to ensure comparability, while avoiding artificial inflation of species richness under short-term fluctuation regimes.

Reviewer's comments 23:

- I would recommend to use appropriate functions to model the fluctuations in T such that the expected behavior per hypothesis is achieved by changing one or few parameter(s) of the same function. Eg., if going after a stochastic model: why not to use white noise vs red noise auto-correlations? e.g., daily temperature fluctuations can be modeled using white noise auto-correlations, while long-term, using red-noise (positive) auto-correlations. This will involve taking the current temperature value in only one step, and the two scenarios could be implemented by changing the value of one parameter. Alternatively, why not to use a deterministic model? E.g., a sine function with different period T depending on the environmental scenario. Wouldn't a deterministic model help making more precise predictions (ie., reduce the effect of stochasticity).

Anyway, the type of selected model needs to be justified.

Authors' responses 23:

Thank you for this thoughtful suggestion. We agree that functions such as sine waves or noise models (e.g., red vs. white noise) offer alternative ways to simulate environmental variability. We chose a nested Gaussian sampling approach because it allows us to systematically vary short-term and long-term thermal fluctuations while maintaining comparable mean temperatures across replicates.

Our approach introduces autocorrelation by design: long-term variation generates the mean for a set of short-term fluctuations, so all short-term steps within a long-term window are drawn around the same baseline. This method captures essential temporal structure without introducing inconsistent baseline temperatures between simulation runs.

To test the robustness of this approach, we implemented a deterministic alternative using sine waves for long-term variation. As described in the Supplementary Fig. 7:

“Model results are robust against alternative and deterministic long-term variation design. Here, instead of sampling from a normal distribution every T_{span} , T_{temp} follows a sine wave with period of 500 time-steps.” (Supplementary Figures, L406-408)

Both the stochastic and deterministic models produced consistent patterns of trait diversification, supporting the robustness of our conclusions.

We opted not to use red noise because it can shift the mean environmental condition across simulation runs, especially near the end of a time series. This would introduce unwanted variation in trait space simply due to run-to-run drift in the background environment, which could obscure comparisons among hypotheses. Our nested design avoids this issue while still allowing us to test the distinct effects of short- vs. long-term variation.

Finally, the stochastic structure better reflects the unpredictable nature of real-world climate fluctuations, particularly for short-term (e.g., daily) changes. Overall, this design allows us to balance biological realism, hypothesis testing precision, and computational consistency.

Reviewer's comments 24:

- There are model decisions taken to ensure "fair competition". What is that? This phrase does not seem biological.

Authors' responses 24:

Thanks for pointing out this issue, we have rephrased it to 'standardized competition' to reduce ambiguity.

Reviewer's comments 25:

- In S5, "min" is written as a function, not as an argument. Please clarify. Furthermore, what is a coefficient for weighting? Also, please explain how the min function works. It seems to take the minimum value from a vector that contains the coefficient for weighting, the density of species j at patch i , and the unity? Thus, p_{compete} can be less or equal to 0.5. However, this is not clear.

Authors' responses 25:

Thank you for pointing out this ambiguity. We have clarified the use of the $\text{min}()$ function in the revised supplementary methods to avoid confusion.

In expression S5, the $\text{min}()$ function returns the smaller of two values:

- The product of a weighting coefficient (c_{weight}) and the population density of species j at patch i ($n_{i,j}$)
- The constant value 1

Mathematically: $\text{min}(c_{\text{weight}} \times n_{i,j}, 1)$

This function ensures that each species' contribution to competitive pressure is capped at 1, preventing highly abundant species from disproportionately dominating the competition calculation. At the same time, rare species (with low densities) contribute proportionally less.

We've clarified this in the revised text as follows:

"In expression S5, n_{spcs} is the number of species, minimum is the function that output the minimum of values within, c_{weight} is the coefficient for weighting and $n_{i,j}$ is the density of species j at patch i . The implementation of minimal function here is to discount rare species that has very few individuals (i.e., when $c_{\text{weight}} \times n_{i,j}$ is less than 1)." (Supplementary Methods, L165-168)

The total weight from all species then modulates the probability of density-dependent reproduction (p_{compete}). As this weight increases (due to more competitors or higher densities), p_{compete} increases but saturates, reflecting empirically observed density-dependent dynamics.

So yes, as you inferred, $p_{compete}$ can indeed be lower than 0.5 in low-competition contexts and approach 1 under strong crowding.

Reviewer's comments 26:

- regarding mortality, it is not clear whether it means that a species either goes extinct or not with given probability; or an individual inside a species dies with the given probability.

Authors' responses 26:

Thank you for this important clarification request. Mortality in our model is applied at the individual level—not at the species level. That is, each individual organism has a fixed probability of dying in each time step. This allows for gradual changes in population size rather than binary species extinction events.

We have clarified this in the revised supplementary methods:

“For simplicity, we assume that mortality does not change with thermal conditions. Therefore, mortality is a fixed probability, p_{mort} , for all species at individual level across the whole simulation.” (Supplementary Methods, L181-182)

This formulation produces more realistic dynamics, where species may persist, decline, or recover depending on the balance between mortality and reproduction, rather than being instantly eliminated due to a single probabilistic event.

Reviewer's comments 27:

- "Because allopatric speciation is much easier to evolve than sympatric speciation". Given the current level of knowledge, allopatric speciation is easier to understand than sympatric speciation. This does not mean that is much easier to evolve. If someone has demonstrated such an statement then add the citation here.

Authors' responses 27:

Thank you for pointing this out. We agree that the original phrasing was imprecise and could be misinterpreted as making a biological claim that is not currently supported by empirical consensus.

Our intention was not to assert that allopatric speciation is inherently easier to evolve, but rather to explain why immigration-based dynamics—analogueous to allopatric speciation—offer practical advantages for our modeling framework.

We have revised the supplementary methods to clarify this:

“In our main model, we simulate the introduction of new species via immigration, analogueous to allopatric speciation. This approach was chosen because it facilitates efficient exploration of the thermal trait space and accelerates the system's convergence to evolutionary steady states, particularly under fluctuating environmental conditions.” (Supplementary Methods, L189-192)

This revised text avoids unsupported evolutionary claims and instead emphasizes the computational and conceptual rationale for our modeling choice.

Reviewer's comments 28:

- Again, it is not clear what an assemblage is. Is it the result of averaging, summing up, or something else, all the patches? If it consists of averaging, then letting offspring migrate might have an impact on the resulting assemblage. Immigration is already implemented by arrivals of random species from the outside. On the other hand, how the variation was changed for the variability hypotheses? this is not clear, and this is crucial for the resulting predictions. Besides, immigration and reproduction dynamics might also differ depending on the variability hypothesis in question. E.g., in the case of daily fluctuations, do moth species undergo reproduction each day? At the moment, modelling decisions seem to suggest that, which would not match the real case study.

Authors' responses 28:

Thank you for raising these critical points. We have revised the supplementary methods to clarify the model structure and terminology.

Assemblage definition:

We now define an “assemblage” as:

“The full set of species with at least one surviving individual across all patches at a given time point.” (Supplementary Methods, L31-32)

This represents the total realized species diversity in the system, not an average or sum across patches. It reflects the collective pool of coexisting thermal strategies at each time step.

Offspring migration and spatial structure:

To maintain demographic connectivity across patches while preserving spatial structure, newborn individuals disperse randomly:

“New-born individuals of each species can freely disperse to any patch. For each offspring, we randomly pick a patch and add one adult to the chosen patch.” (Supplementary Methods, L185-186)

This setup allows for species persistence and local rescue effects without conflating dispersal with external immigration.

Temporal scaling and variability manipulation:

As noted, our model uses abstract time steps that represent a flexible timescale of population processes, not literal days:

“Our model is a heuristic framework that captures environmental fluctuations across multiple time scales, but it is not intended for direct translation of a single time step into a real-world unit such as a day.” (Supplementary Methods, L32-34)

This design avoids mismatches between real-life moth life cycles and model iteration speed. We’ve clarified that reproduction and mortality occur per time step to facilitate comparisons across different fluctuation regimes.

Environmental variability modeling:

To manipulate short- and long-term variability independently while keeping the mean temperature constant, we use a two-tiered sampling approach:

“The temporary average is based on mean ambient temperature and long-term thermal variability, $T_{temp} \sim \text{Norm}(\mu = T_{mean}, \sigma = T_{sd,long})$, while current temperature is based on the sampled temporary average and short-term thermal variability, $T_{current} \sim \text{Norm}(\mu = T_{temp}, \sigma = T_{sd,short})$.” (Supplementary Methods, L40-42)

This nested design ensures that our variability hypotheses are implemented in a consistent, interpretable way while isolating the effects of short-term vs. long-term environmental fluctuations.

Reviewer's comments 1:

REVIEWER COMMENTS

Reviewer #1 (Remarks to the Author):

Thanks for the thorough set of responses and edits made in response to the previous round of reviews. The manuscript appears significantly stronger for it, and I'm largely satisfied with the new draft.

The one area that still needs attention is the Discussion. In response to my prior comments, you added sections on shivering endothermy and on microclimates. This is good overall, but the writing is odd here. There are two paragraphs that both start with almost identical topic sentences "Most sampled moths are nocturnal fliers that elevate thoracic temperatures to ~33–38 °C via shivering endothermy." Some of the other information in the paragraph also is redundant. I would suggest either (1) combining the paragraph into one, so that you're focusing on two related issues that may alter the thermal experience of moths (although I can see that this would be a giant paragraph), or (2) structuring the paragraphs so that they're more distinct from the beginning, one on flight- or shivering-generated heat production and the other on microclimatic choices. These really are different, as one comes from within the moths and the other from the outside, based on the choices they make about where to go and when to be active.

Authors' responses 1:

We appreciate your careful guidance on the Discussion's structure. In the revised manuscript we resolved the redundancy by consolidating the two overlapping paragraphs into a single, integrated paragraph.

We now open with a succinct sentence on shivering endothermy to acknowledge its role in buffering operative temperatures during flight, and then pivot to emphasize that selection likely acts most strongly during inactive periods, when individuals—especially immobile stages—experience fine-scale microclimates.

For now, we chose the single integrated paragraph to keep the narrative concise and to foreground how internal and external thermal contexts jointly

inform selection on thermal traits. We hope this addresses your concern and improves readability, and we remain grateful for your helpful suggestion.

new discussion paragraph:

“Most sampled moths are nocturnal fliers that rely on shivering endothermy to elevate thoracic temperatures to approximately 33–38 °C³³, thereby buffering operative temperature fluctuations during activity. However, selective pressures on thermal physiology are also likely to act during periods of inactivity. Although adult moths typically seek shaded refugia during daylight hours, immobile life stages remain exposed to fine-scale microclimatic gradients shaped by topography and solar input. Microclimatic thermal heterogeneity exhibits significant three-way interactions among region, diel period, and elevation. Across multiple regions and time periods, high-elevation environments consistently show greater thermal heterogeneity³⁴, with daytime variability exceeding that of nighttime. Previous studies further indicate that in mountainous systems, heterogeneity is strongly shaped by topographic roughness, canopy architecture, and spatiotemporal variation in precipitation³⁴. In the montane cloud forests of Taiwan, fog and low cloud layers exert powerful radiative control over the diurnal temperature range (DTR), producing nonlinear elevational gradients and marked discontinuities at mid- to high elevations³⁵. These discontinuities align with zones of persistent fog and highlight the complex spatial structure of thermal regimes in montane ecosystems. Within this framework, environmental favorability—defined by the confluence of benign temperatures and high primary productivity—emerges as a principal driver of thermal trait diversity. Warm, productive environments support both thermal specialists and generalists, expanding trait hypervolume and increasing assemblage-level variability. By contrast, although high-elevation habitats exhibit substantial microclimatic variation, their overall thermal severity imposes strong environmental filtering that compresses trait distributions. Greater DTR may elevate CTmax in sedentary stages and increase average TTrange, yet this occurs without corresponding expansion in trait hypervolume. These findings support the idea that even short-term fluctuations, occurring on timescales shorter than species’ generation lengths, can shape evolutionary outcomes distinct from those driven by long-term means. However, local environmental conditions—averaged over biologically relevant timescales—may exert stronger selective pressures than those reflected in long-term climatic averages alone³⁶.” (L317-339)

Reviewer's comments 2:

Reviewer #2 (Remarks to the Author):

The authors did a good job addressing my concerns and improving the communication of the model.

I have a minor comment:

- On the use of fixed immigration / mutation rates across environmental scenarios: One needs to be cautious with this assumption, since it can lead to an overestimation of the adaptive response of populations. This is particularly relevant under scenarios of "fast" (relative to generation time) change. Thus, for instance the model could predict persistence when in reality extinction is observed. Though to the purpose of the study such model assumption could be justified, please add a remark to make readers aware of this.

Authors' responses 2:

We agree and have added a sentence in the Discussion acknowledging this limitation and noting it as a direction for future work.

Relevant new Discussion:

“To isolate environmental effects on competition, we held immigration/mutation rate constant across scenarios; although this can potentially overestimate adaptive responses under rapid change, it offers a clear baseline, and a natural next step is to relax this assumption to test the robustness of persistence predictions.” (L365-368)

Reviewer's comments 3:

Reviewer #2 (Remarks on code availability):

The code is written in C language. It has a read-me file with directions to the source and data files. Potential future users will need to find their way by navigating and experiencing the code. Though the source code is commented, it might not be accessible for users without knowledge on C programming language.

Authors' responses 3:

We thank the reviewer for these helpful comments and for prompting us to improve accessibility. In the revised repository, we (i) rewrote the README to

consolidate all documentation, and (ii) added copy-and-paste commands in the README so users can reproduce results without editing C code. Under Simulations section, we now provide explicit compile and run examples:

(a) `gcc -O2 rtsc_nspc.c -lm -o rtsc_nspc`: compiles the source code into an executable in one step, no code changes are required.

(b) `./rtsc_nspc mean=18.0 svar=2.5 lvar=0.0 &`: runs the program while setting temperature parameters directly from the command line, so users need not open or edit the code. Here, *mean*, *svar*, and *lvar* denote mean ambient temperature, short-term variability, and long-term variability, respectively. These values can be adjusted to reproduce or explore different scenarios, even without any background in C.

We also note the bundled dependency (dSFMT-src-2.3.3, Fast Mersenne Twister) and provide its upstream source. Taken together, these changes are intended to make the code runnable by users without proficiency in C. For readers who wish to modify or extend the model, the original, extensively commented source remains available.